# Genomic diversity across the *Rickettsia* and '*Candidatus* Megaira' genera and proposal of genus status for the Torix group

Helen R. Davison[1], Jack Pilgrim[1], Nicky Wybouw[2], Joseph Parker[3], Stacy Pirro[4], Simon Hunter-Barnett[1], Paul M. Campbell[1,5], Frances Blow[1,6], Alistair C. Darby[1], Gregory D. D. Hurst[1] & Stefanos Siozios[1✉]

Members of the bacterial genus *Rickettsia* were originally identified as causative agents of vector-borne diseases in mammals. However, many *Rickettsia* species are arthropod symbionts and close relatives of '*Candidatus* Megaira', which are symbiotic associates of microeukaryotes. Here, we clarify the evolutionary relationships between these organisms by assembling 26 genomes of *Rickettsia* species from understudied groups, including the Torix group, and two genomes of '*Ca*. Megaira' from various insects and microeukaryotes. Our analyses of the new genomes, in comparison with previously described ones, indicate that the accessory genome diversity and broad host range of Torix *Rickettsia* are comparable to those of all other *Rickettsia* combined. Therefore, the Torix clade may play unrecognized roles in invertebrate biology and physiology. We argue this clade should be given its own genus status, for which we propose the name '*Candidatus* Tisiphia'.

[1] Institute of Infection, Veterinary and Ecological sciences, University of Liverpool, Liverpool L69 7ZB, UK. [2] Terrestrial Ecology Unit, Department of Biology, Faculty of Sciences, Ghent University, Ghent, Belgium. [3] Division of Biology and Biological Engineering, California Institute of Technology, 1200 E California Boulevard, Pasadena, CA 91125, USA. [4] Iridian Genomes, Bethesda, MD, USA. [5] School of Health and Life Sciences, Faculty of Biology Medicine and Health, the University of Manchester, Manchester, UK. [6] Center for Genomics and Systems Biology, Department of Biology, New York University, New York, NY, USA. ✉email: siozioss@liverpool.ac.uk

Symbiotic bacteria are vital to the function of most living eukaryotes, including microeukaryotes, fungi, plants, and animals[1–4]. The symbioses formed are often functionally important to the host with effects ranging from mutualistic to detrimental. Mutualistic symbionts may provide benefits through the biosynthesis of metabolites, or by protecting their hosts against pathogens and parasitoids[5,6]. Parasitic symbionts can be detrimental to the host due to resource exploitation or through reproductive manipulation that favours its own transmission over the host's[7,8]. Across these different symbiotic relationships, symbionts are often important determinants of host ecology and evolution.

The *Rickettsiales* (Alphaproteobacteria) represent an order of largely obligate intracellular bacteria that form symbioses with a variety of eukaryotes[9]. *Deianiraea*, an extracellular parasite of *Paramecium*, is the one known exception[10]. Within *Rickettsiales*, the family *Rickettsiaceae* represent a diverse collection of bacteria that infect a wide range of eukaryotic hosts and can act as symbionts, parasites, and pathogens. Perhaps the best-known clade of *Rickettsiaceae* is the genus *Rickettsia*, which was initially described as the cause of spotted fever and other rickettsioses in vertebrates that are transmitted by ticks, lice, fleas, and mites[11].

*Rickettsia* have been increasingly recognised as heritable arthropod symbionts. Since the description of a maternally inherited male-killer in ladybirds[12], we now know that heritable *Rickettsia* are common in arthropods[13,14]. Further, *Rickettsia*-host symbioses are diverse, with different symbionts being capable of reproductive manipulation, nutritional and protective symbiosis, as well as influencing thermotolerance and pesticide susceptibility[15–21].

Our understanding of the evolution and diversity of the genus *Rickettsia* and its allies has increased in recent years, with the taxonomy of *Rickettsiaceae* developing as more data becomes available[14,22]. Weinert et al.[14] loosely defined 13 different groups of *Rickettsia* based on 16 S rRNA phylogeny, which showed two early branching clades that appeared genetically distant from other members of the genus. One of these was a symbiont of *Hydra* and designated as Hydra group *Rickettsia*, which has since been assigned its own genus status, 'Candidatus Megaira'[23]. 'Ca. Megaira' forms a related clade to *Rickettsia* and is found in ciliates, amoebae, chlorophyte and streptophyte algae, and cnidarians[24]. Members of this clade are found in hosts from aquatic, marine and soil habitats which include model organisms (e.g., *Paramecium*, *Volvox*) and economically important vertebrate parasites (e.g., *Ichthyophthirius multifiliis*, the ciliate that causes white spot disease in fish)[24]. Whilst symbioses between 'Ca. Megaira' and microeukaryotes are pervasive, there is no publicly available complete genome and the impact of these symbioses on the host are poorly understood.

A second early branching clade was described from *Torix tagoi* leeches and is commonly coined Torix group *Rickettsia*[25]. Symbionts in the Torix clade have since been found in a wide range of invertebrate hosts from midges to freshwater snails to fish-parasitic amoeba[13]. The documented diversity of hosts is wider than other *Rickettsia* groups, which are to date only found in arthropods and their associated vertebrate or plant hosts[14]. Torix clade *Rickettsia* are known to be heritable symbionts, but their impact on host biology is poorly understood, despite the economic and medical importance of several hosts (inc. bed bugs, black flies, and biting midges). Rare studies have described the potential effects on the host, which include larger body size in leeches[25]; a small negative effect on growth rate and reproduction in bed bugs[26]; and an association with parthenogenesis in *Empoasca* Leafhoppers[27].

Current data suggest an emerging macroevolutionary scenario where the members of the *Rickettsia* clade originated as symbionts of microeukaryotes, before diversifying to infect invertebrates[23,28,29]. Many symbionts belonging to the *Rickettsiaceae* (e.g., 'Ca. Megaira', 'Candidatus Trichorickettsia', 'Candidatus Phycorickettsia', 'Candidatus Sarmatiella' and 'Candidatus Gigarickettsia') circulate in a variety of microeukaryotes[23,30–33]. The Torix group *Rickettsia* retained a broad range of hosts from microeukaryotes to arthropods[13]. The remaining members of the genus *Rickettsia* evolved to be arthropod heritable symbionts and vector-borne pathogens[14,34]. However, a lack of genomic and functional information for symbiotic clades limits our understanding of evolutionary transitions within *Rickettsia* and its related groups. No 'Ca. Megaira' genome sequences are currently publicly available and of the 165 *Rickettsia* genome assemblies available on the NCBI (as of 29/04/21), only two derive from the Torix clade and these are both draft genomes. In addition, dedicated heritable symbiont clades of *Rickettsia*, such as the Rhyzobius group, have no available genomic data, and there is a single representative for the Adalia clade. Despite the likelihood that heritable symbiosis with microeukaryotes and invertebrates was the ancestral state for this group of intracellular bacteria, available genomic resources are heavily skewed towards pathogens of vertebrates.

In this study we establish a richer base of genomic information for heritable symbionts *Rickettsia* and 'Ca. Megaira', then use these resources to clarify the evolution of these groups. We broaden available genomic data through a combination of targeted sequencing of strains without complete genomes, and metagenomic assembly of *Rickettsia* strains from arthropod genome projects. We report the first closed circular genome of a 'Ca. Megaira' symbiont from a streptophyte alga (*Mesostigma viride*) and provide a draft genome for a second 'Ca. Megaira' from a chlorophyte (*Carteria cerasiformis*). In addition, we present the complete genomes of two Torix *Rickettsia* from a midge (*Culicoides impunctatus*) and a bed bug (*Cimex lectularius*) as well as a draft genome for *Rickettsia* from a tsetse fly (*Glossina morsitans submorsitans*, an important vector species), and a new strain from a spider mite (*Bryobia graminum*). A metagenomic approach established a further 22 draft genomes for insect symbiotic strains, including previously unsequenced Rhyzobius and Meloidae group draft genomes. We utilize these to conduct pangenomic, phylogenomic, and metabolic analyses of our extracted genome assemblies, with comparisons to existing *Rickettsia*.

## Results and discussion

We have expanded the available genomic data for several *Rickettsia* groups through a combination of draft and complete genome assembly. This includes an eight-fold increase in available Torix-group genomes, and genomes for previously unsequenced Meloidae and Rhyzobius groups. We further report initial reference genomes for 'Ca. Megaira'.

**Complete and closed reference genomes for Torix *Rickettsia* and 'Ca. Megaira'.** The use of long-read sequencing technologies produced complete genomes for two subclades of the Torix group limoniae (RiCimp) and leech (RiClec). Sequencing depth of the *Rickettsia* genomes from *C. impunctatus* (RiCimp) and *C. lectularius* (RiClec) were 18X and 52X, respectively. The RiCimp genome provides evidence of plasmids in the Torix group (pRiCimp001 and pRiCimp002) (Table 1). Notably, the two plasmids share more similarities between them than to other *Rickettsia* plasmids. However, both plasmids contain distant homologs of the DnaA_N domain-containing proteins previously found in other *Rickettsia* plasmids[35]. In addition, only two components of the type IV conjugative transfer system known as RAGEs

**Table 1 Summary of the closed 'Ca. Megaira' and Torix *Rickettsia* genomes completed in this project.**

| Group | 'Ca. Megaira' | Torix *Rickettsia* | Torix *Rickettsia* |
|---|---|---|---|
| Strain Name | MegNIES296 | RiCimp | RiClec |
| Symbiont genome accession | GCA_020410825.1 | GCA_020410785.1 | GCA_020410805.1 |
| Host | Mesostigma viride NIES-296 | *Culicoides impunctatus* | Cimex lectularius |
| Raw reads accession | SRR8439255, SRX5120346 | SRR16018514, SRR16018513 | SRR16018512, SRR16018511 |
| Total nucleotides | 1,532,409 | 1,566,468 | 1,611,726 |
| Chromosome size (bp) | 1,448,425 | 1,469,631 | 1, 611,726 |
| Plasmids | 1 (83,984 bp) | 2 (77550 bp + 19287 bp) | None |
| GC content (%) | 33.9 | 32.9 | 32.8 |
| Number of CDS | 1,359 | 1,397 | 1,544 |
| Avg. CDS length (bp) | 998 | 900 | 874 |
| Coding density (%) | 88.5 | 86 | 84 |
| rRNAs | 3 | 3 | 3 |
| tRNAs | 34 | 34 | 35 |

(*Rickettsiales* Amplified Genetic Elements)[36] were present on the plasmids including homologs of the proteins TrwB/TraD and TraA/MobA. The majority of the RAGE elements including both the F-like (*tra*) and P-like type IV components have been incorporated in the main chromosome. The presence of RAGE elements, alongside the fact conjugation apparatuses have narrow host-ranges[37], suggest horizontal transfer of these plasmids is likely within the *Rickettsiaceae* and could occur between Torix and the main *Rickettsia* clade, considering co-infections of these genera have been noted previously[38,39]. We additionally assembled a complete closed reference genome of 'Ca. Megaira' from *Mesostigma viride* (MegNEIS296) from previously published genome sequencing efforts. Likewise, MegNEIS296 genome contains a plasmid which bears features of other *Rickettsia* plasmids including the presence of a tra conjugative element and the presence of two DnaA_N-like protein paralogs.

General features of both genomes are consistent with previous genomic studies of the Torix group (Table 1). A single full set of rRNAs (16 S, 5 S and 23 S) and a GC content of ~33% was observed. Notably, the two complete Torix group genomes show a distinct lack of synteny (Supplementary Fig. 1), a genomic feature that is compatible with our phylogenetic analyses that placed these two lineages in different subclades (leech/limoniae) (Fig. 1 and Supplementary Fig. 3). Gene order breakdown due to intragenomic recombination has been previously associated with the expansion of mobile genetic elements in both *Rickettsia*[40] and *Wolbachia*[41], another member of the *Rickettsiales*. Both RiCimp and RiClec genomes predicted to encode for a high number of transposable elements with circa 96 and 119 annotated putative transposases, respectively. This expansion of transposable elements along with their phylogenetic distance is likely responsible for the extreme synteny breakdown between RiCimp and RiClec. Of note within the closed reference genomes MegNEIS296 and RiCimp is the presence of a putative non-ribosomal peptide synthetase (NRPS) and a hybrid non-ribosomal peptide/polyketide synthetase (NRPS/PKS) respectively (Supplementary Fig. 2). Although, the exact products of these putative pathways are uncertain, in silico prediction by Norine suggests some similarity with both cytotoxic and antimicrobial peptides hinting at a potential defensive role (Supplementary Fig. 2). Further homology comparison with other taxa did not provide links with any specific functions or phenotypes. Previously, an unrelated hybrid NRPS/PKS cluster has been reported in *Rickettsia buchneri* on a mobile genetic element, providing potential routes for horizontal transmission[42]. The strongest blastp hits of MegNEIS296 NRPS proteins occur in *Cyanobacteria* (Supplementary Fig. 2)[42]. In addition, putative toxin-antitoxin systems similar to one associated with cytoplasmic incompatibility in *Wolbachia* have recently been

observed on the plasmid of *Rickettsia felis* in a parthenogenetic booklouse[35]. Toxin-antitoxin systems are thought to be part of an extensive bacterial mobilome network associated with reproductive parasitism[43]. A BLAST search found a very similar protein in Oopac6 to the putative large pLbAR toxin found in *R. felis* (88% aa identity), and a more distantly related protein in the *C. impunctatus* plasmid (25% aa identity).

**Sequencing and de novo assembly of other *Rickettsia* and 'Ca. Megaira' genomes.** Our direct sequencing efforts enabled assembly of draft genomes for a second 'Ca. Megaira' strain from the alga *Carteria cerasiformis*, and for *Rickettsia* associated with tsetse flies and *Bryobia* spider mites. The *Rickettsia* genome retrieved from a wild caught Tsetse fly, RiTSETSE, is a potentially chimeric assembly of closely related Transitional group *Rickettsia*. We identified an excess of 3584 biallelic sites (including 3369 SNPs and 215 indels) when the raw Illumina reads were mapped back to the assembly. High read depth of 104X indicate that this could be a symbiotic association, reflecting previous observations in Tsetse fly cells[44]. However, there is a possibility that RiTSETSE is not a heritable symbiont but comes from transient infection from a recent blood meal.

From the SRA accessions, the metagenomic pipeline extracted 29 full symbiont genomes for *Rickettsiales* across 24 host species. Five of 29 were identified as *Wolbachia* and discarded from further analysis, one was a *Rickettsia* discarded for low quality, and another was a previously assembled Torix *Rickettsia*, RiCNE[45]. Thus, 22 high quality *Rickettsia* metagenomes were obtained from 21 host species. One beetle (SRR6004191) carried coinfecting *Rickettsia* Lappe3 and Lappe4 (Table 2). The high-quality *Rickettsia* genomes covered the Belli, Torix, Transitional, Rhyzobius, Meloidae and Spotted Fever Groups (Table 2 and Supplementary Data 1).

Beetles, particularly rove beetle (*Staphylinidae*) species, appear in this study as a possible hotspot of *Rickettsia* infection. *Rickettsia* has historically been commonly associated with beetles, including ladybird beetles (*Adalia bipunctata*), diving beetles (*Deronectes sp.*) and bark beetles (*Scolytinae*)[14,17,34,46,47]. Though a plausible and likely hotspot, this observation needs be approached with caution as this could be an artefact of skewed sampling efforts.

**Phylogenomic analyses and taxonomic placement of assembled genomes.** The phylogeny and network illustrate the distance of Torix from 'Ca. Megaira' and other *Rickettsia*, along with an extremely high level of within-group diversity in Torix compared

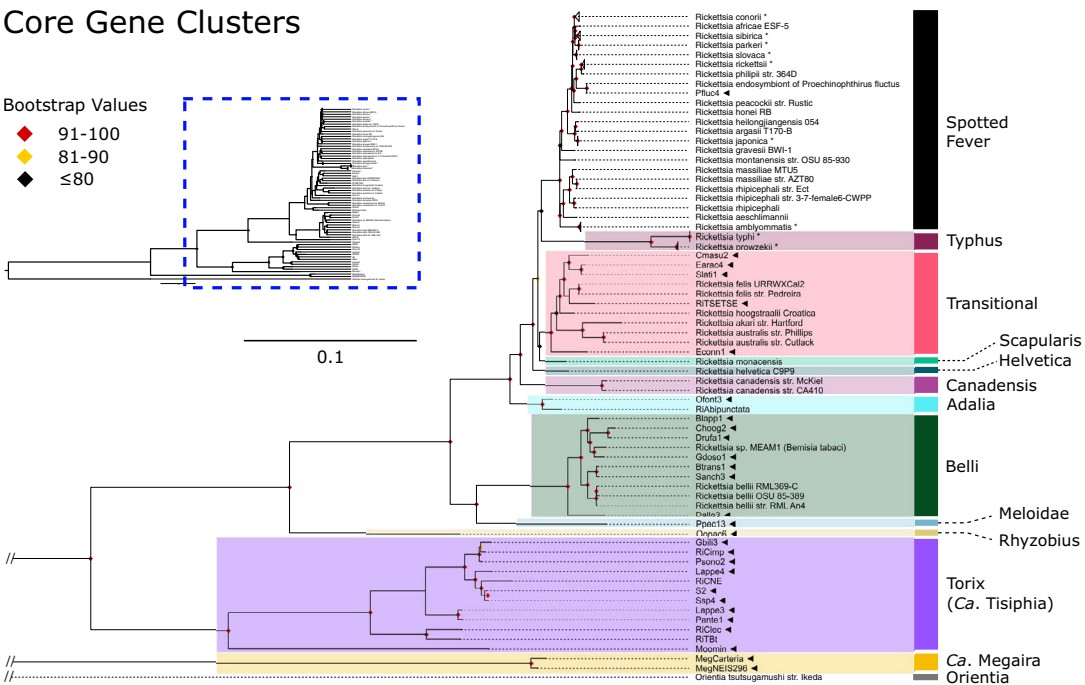

**Fig. 1 Genome wide phylogeny of *Rickettsia* and '*Ca*. Megaira'.** Maximum likelihood (ML) phylogeny of *Rickettsia* and '*Ca*. Megaira' constructed from 74 core gene clusters extracted from the pangenome. New genomes are indicated by ◄ and bootstrap values based on 1000 replicates are indicated with coloured diamonds (red = 91–100, yellow = 81–90, black < = 80). New complete genomes are: RiCimp, RiClec and MegNEIS296. Asterisks indicate collapsed monophyletic branches and "//" represent breaks in the branch. Accessions used are provided in Supplementary Data 1.

to any other group (Fig. 1 and Supplementary Fig. 3). No significant discordance was detected between the core and ribosomal phylogenies. The phylogenies generated using core genomes are consistent with previously identified *Rickettsia* and host associations using more limited genetic markers[13,14,48,49]. For instance, Pfluc4 from *Proechinophthirus fluctus* lice is grouped on the same branch as a previously sequenced *Rickettsia* from a different individual of *P. fluctus*[48]. The following groups were identified in the 22 genomes assembled from the SRA screening: 4 Transitional, 1 Spotted Fever, 1 Adalia, 8 Belli and 7 Torix limoniae. Targeted sequences were confirmed as: Torix limoniae (RiCimp), Torix leech (RiClec), Transitional (RiTSETSE), '*Ca*. Megaira' (MegCarteria and MegNEIS296), and a deeply diverging Torix clade provisionally named Moomin (Moomin) (Table 2, Fig. 1, Supplementary Fig. 3 and 4). The extracted Torix genomes include one double infection giving a total of 10 new genomes across 9 potential host species. The double infection is found within the rove beetle *Labidopullus appendiculatus*, forming two distinct lineages, Lappe3 and Lappe4 (Fig. 1 and Supplementary Fig. 3).

We also report a putative Rhyzobius group *Rickettsia* genomes extracted from the staphylinid beetle *Oxypoda opaca* (Oopac6) and Meloidae group *Rickettsia* from the firefly *Pyrocoelia pectoralis* (Ppec13). They have high completeness, low contamination, and consistently group away from the other draft and completed genomes (Figs. 1, 2, and Supplementary Data 1). MLST analyses demonstrate that these bacteria are most like the Rhyzobius and Meloidae groups described by Weinert et al.[14] (Supplementary Fig. 4). Phylogenies of Oopac6 and Ppec13 suggest that Rhyzobius sits as sister group to all other *Rickettsia* groups, and Meloidae is more closely associated with Belli (Fig. 1, Supplementary Fig. 3–5). Further genome construction will help clarify this taxon and its relationship to the rest of the *Rickettsiaceae*. The sequencing data for the wasp, *Diachasma*

*alloeum*, used here has previously been described to contain a pseudogenised nuclear insert of *Rickettsia* material, but not a complete *Rickettsia* genome[50]. The construction of a full, non-pseudogenised genome with higher read depth than the insect contigs, low contamination (0.95%) and high completion (93.13%) suggests that these reads likely represent a viable *Rickettsia* infection in *D. alloeum*. However, these data do not exclude the presence of an additional nuclear insert. It is possible for a whole symbiont genome to be incorporated into the host's DNA like in the case of *Wolbachia*[51], or the partial inserts of '*Ca*. Megaira' genomes in the *Volvox carteri* genome[52]. The presence of both the insert and symbiont need confirmation through appropriate microscopy methods.

Recombination is low within the core genomes of *Rickettsia* and '*Ca*. Megaira' but may occur between closely related clades that are not investigated here. Across all genomes, the PHI score is significant in 6 of the 74 core gene clusters, suggesting putative recombination events. However, it is reasonable to assume that most of these may be a result of systematic error due to the divergent evolutionary processes at work across *Rickettsia* genomes. Patterns of recombination can occur by chance rather than driven by evolution which cannot be differentiated by current phylogenetic methods[53]. The function of each respective cluster can be found in Supplementary Data 1.

**Gene content, pangenome and metabolic analysis.** Across all genomes used in the gene content comparison analysis (Supplementary Fig. 6), Anvi'o identified only 208 core gene clusters of which 74 are represented by single-copy genes. It is particularly evident the large size of the accessory genome across the main *Rickettsia* and the Torix clades. Out of the 2470 predicted ortholog clusters for the Torix clade 1296 (52.5%) are uniquely found among the Torix genomes, while for *Rickettsia* 2460 unique ortholog clusters were predicted from a total of 3811 (64.5%)

**Table 2 Summary of draft genomes generated during the current project and their associated hosts. Full metadata including CheckM completeness scores and levels of contamination can be found in Supplementary Data 1.**

| Strain | Symbiotic bacteria assembly accession | Group | Number of contigs | Total length (bp) | Host name | Host Order |
|---|---|---|---|---|---|---|
| Blapp1 | GCA_0204404495.1 | Belli | 171 | 1266633 | Bembidion lapponicum | Coleoptera |
| Btrans1 | GCA_0204404375.1 | Belli | 241 | 1417452 | Bembidion nr. transversale OSAC:DRMaddison DNA3205 | Coleoptera |
| Choog2 | GCA_0204404365.1 | Belli | 16 | 1357829 | Columbicola hoogstraali | Phthiraptera |
| Cmasu2 | GCA_0204404525.1 | Transitional | 196 | 1295004 | Ceroptres masudai | Hymenoptera |
| Dallo3 | GCA_0204404485.1 | Belli | 196 | 990679 | Diachasma alloeum | Hymenoptera |
| Drufa1 | GCA_0204404445.1 | Belli | 14 | 1364611 | Degeeriella rufa | Phthiraptera |
| Earac4 | GCA_020881375.1 | Transitional | 96 | 1350066 | Ecitomorpha arachnoides | Coleoptera |
| Econn1 | GCA_020881315.1 | Transitional | 238 | 1070326 | Eriopis connexa | Coleoptera |
| Gbili3 | | Torix limoniae ('Ca. Tisiphia') | 171 | 1188102 | Gnoriste bilineata | Diptera |
| Gdoso1 | GCA_020881275.1 | Belli | 34 | 1420758 | Graphium doson | Lepidoptera |
| Lappe3 | GCA_020881245.1 | Torix limoniae ('Ca. Tisiphia') | 122 | 1368980 | Labidopullus appendiculatus | Coleoptera |
| Lappe4 | GCA_020881075.1 | Torix limoniae ('Ca. Tisiphia') | 154 | 1332357 | Labidopullus appendiculatus | Coleoptera |
| MegCarteria | GCA_020881215.1 | 'Ca. Megaira' | 72 | 1298707 | Carteria cerasiformis | Chlamydomonadales |
| Ofont3 | GCA_0204404465.1 | Adalia | 91 | 1529137 | Omalisus fontisbellaquei | Coleoptera |
| Oopac6 | GCA_020881235.1 | Rhyzobius | 181 | 1497231 | Oxypoda opaca | Coleoptera |
| Pante1 | GCA_020881195.1 | Torix limoniae ('Ca. Tisiphia') | 70 | 1472610 | Pseudomimeciton antennatum | Coleoptera |
| Pfluc4 | GCA_0204404545.1 | Spotted Fever | 7 | 1251895 | Proechinophthirus fluctus | Phthiraptera |
| Ppec13 | GCA_0204404425.1 | Belli | 90 | 1426047 | Pyrocoelia pectoralis | Coleoptera |
| Psono2 | GCA_020881175.1 | Torix limoniae 'Ca. Tisiphia' | 163 | 1492063 | Platyusa sonomae | Coleoptera |
| RiTSETSE | GCA_020881295.1 | Transitional | 172 | 1451997 | Glossina morsitans submorsitans | Diptera |
| S2 | GCA_0204404555.1 | Torix limoniae ('Ca. Tisiphia') | 103 | 1251484 | Sericostoma | Trichoptera |
| Sanch3 | GCA_020881115.1 | Belli | 181 | 1487154 | Stiretrus anchorago | Hemiptera |
| Slati1 | GCA_020881155.1 | Transitional | 109 | 1301763 | Sceptobius lativentris | Coleoptera |
| Ssp4 | GCA_0204404565.1 | Torix limoniae ('Ca. Tisiphia') | 87 | 1231013 | Sericostoma sp. HW-2014 | Trichoptera |
| Moomin | GCA_020881085.1 | Torix moomin ('Ca. Tisiphia') | 204 | 1137559 | Bryobia graminum | Trombidiformes |

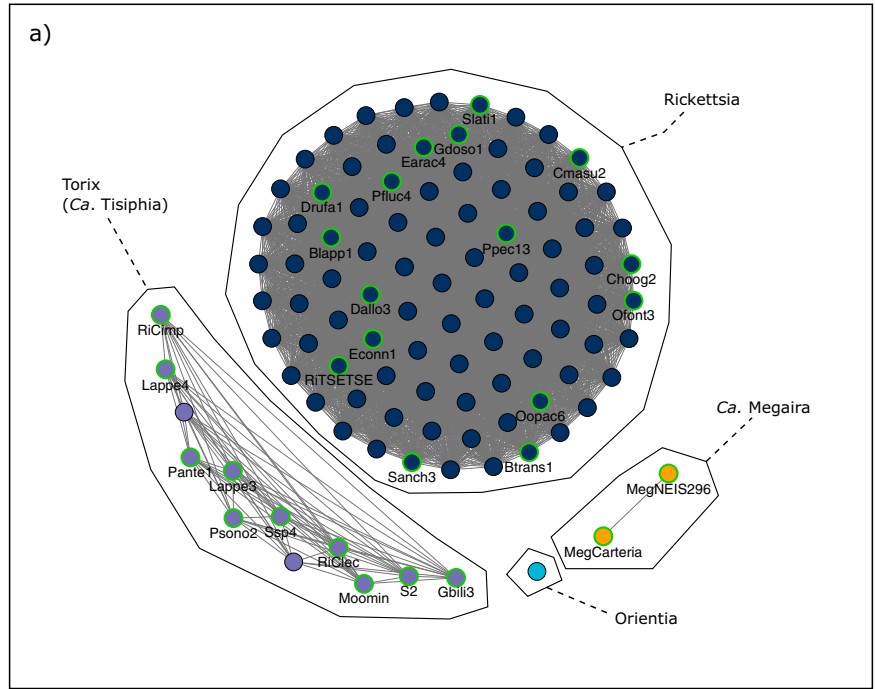

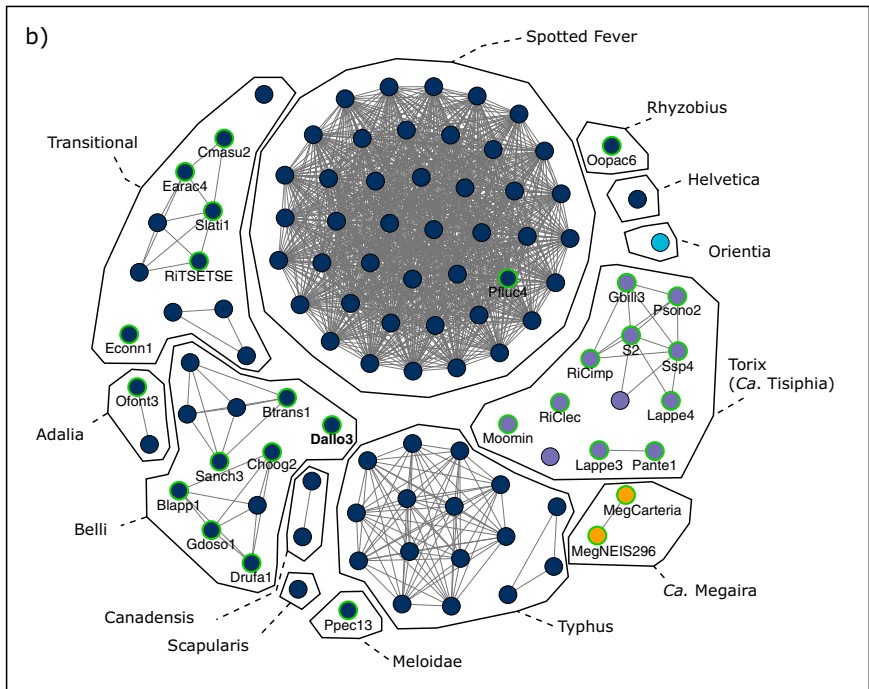

**Fig. 2 Genus and species level clustering across *Rickettsia* and '*Ca*. Megaira'.** Frutcherman Reingold networks of pairwise (**a**) Average Amino Acid Identity (AAI) with edge weights >65% similarity and (**b**) Average Nucleotide Identity (ANI) with edge weights >95% similarity across all genomes. AAI and ANI illustrate genus and species boundaries, respectively. The 13 current cluster names are annotated over the 23 species clusters found in the ANI network. New genomes are named and have a green outline. Node fill colours indicate *Rickettsia* (Dark blue), '*Ca*. Megaira' (orange), Torix/ '*Ca*. Tisiphia' (purple), *Orientia* outgroup (light blue). Source data are provided in Source Data.

(Fig. 3). However, if we account for the number of genomes available in each clade then Torix shows higher rates of gene cluster and unique gene clusters accumulation with each additional genome (Fig. 4). Our results indicate that the main *Rickettsia* clade and especially the Torix clade, seem to have a high degree of genome diversity, suggesting a wider repertoire of genes and potentially greater rates of gene turnover. As expected, the more genomes that are included in analyses, the smaller the core genome extracted. However, gene content analysis results of

increasingly diverged genomes should be always interpreted with caution as true homology relationship between genes/proteins might get obscured by their sequence divergence.

Torix is a distinctly separate clade sharing less than 65% AAI similarity to any *Rickettsia* or '*Ca*. Megaira' genomes (Fig. 2). It contains at least five species-level clusters with >95% ANI similarity that reflect its highly diverse niche in the environment (Fig. 2)[13,54,55]. With only two examples, the true diversity of '*Ca*. Megaira' is underestimated here. Overall, our results indicate that

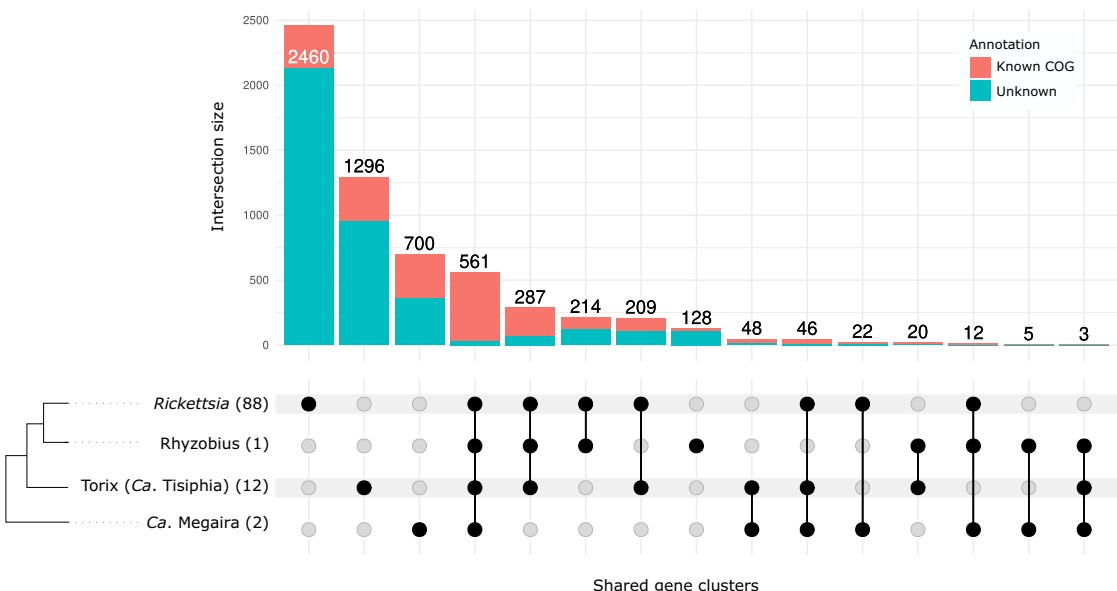

**Fig. 3 Gene content comparison.** Shared and unique gene clusters across genus putative genus clusters *Rickettsia*, Rhyzobius, Torix and 'Ca. Megaira' as suggested by GTDB-tk. Vertical coloured bars represent the size of intersections (the number of shared gene clusters) between genomes in descending order with known COG functions displayed in coral and unknown in blue. Black dots mean the cluster is present and connected dots represent gene clusters that are present across groups. Numbers in parenthesis represent the number of genomes used in the analysis. Source data are provided in Source Data.

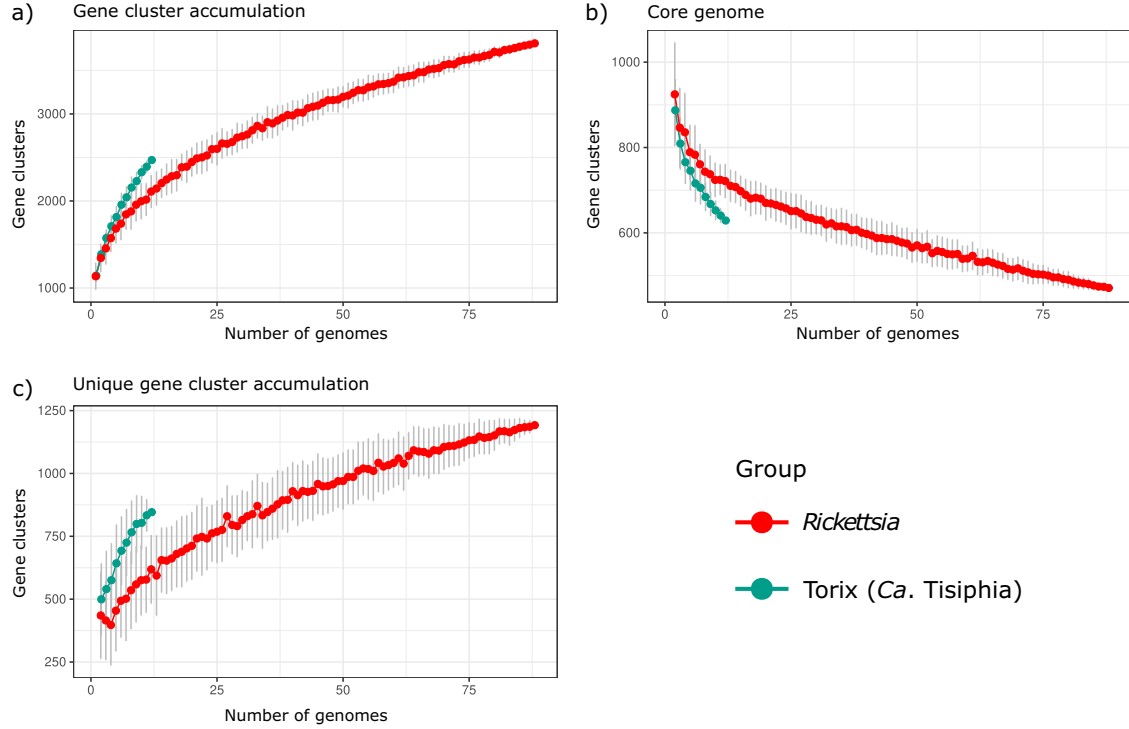

**Fig. 4 Gene cluster accumulation analysis. a** Pangenome accumulation curves. **b** Core genome accumulation curves. **c** The unique genome of *Rickettsia* (red) and Torix (turquoise) clades as a function of the number of genomes sequenced. Each point represents the mean value while error bars represent ± standard deviation based on 100 permutations. Source data are provided in Source Data.

higher genomic plasticity within Torix clade in terms of gene content compared to *Rickettsia*.

We also investigated whether Torix and *Rickettsia* clades are enriched for particular COGs (Supplementary Data 1). Among the most highly enriched genes in Torix clade were genes encoding for invasion associated proteins like the

exopolysaccharide synthesis protein ExoD (COG3932) and the invasion associated protein IalB (COG5342), a carbonic anhydrase (COG0288) and a Chloramphenicol resistance associated protein (COG3896). Both carbonic anhydrase and ExoD homologs has been already reported in Torix clade[45] and our results here further support their important role in Torix biology. ExoD

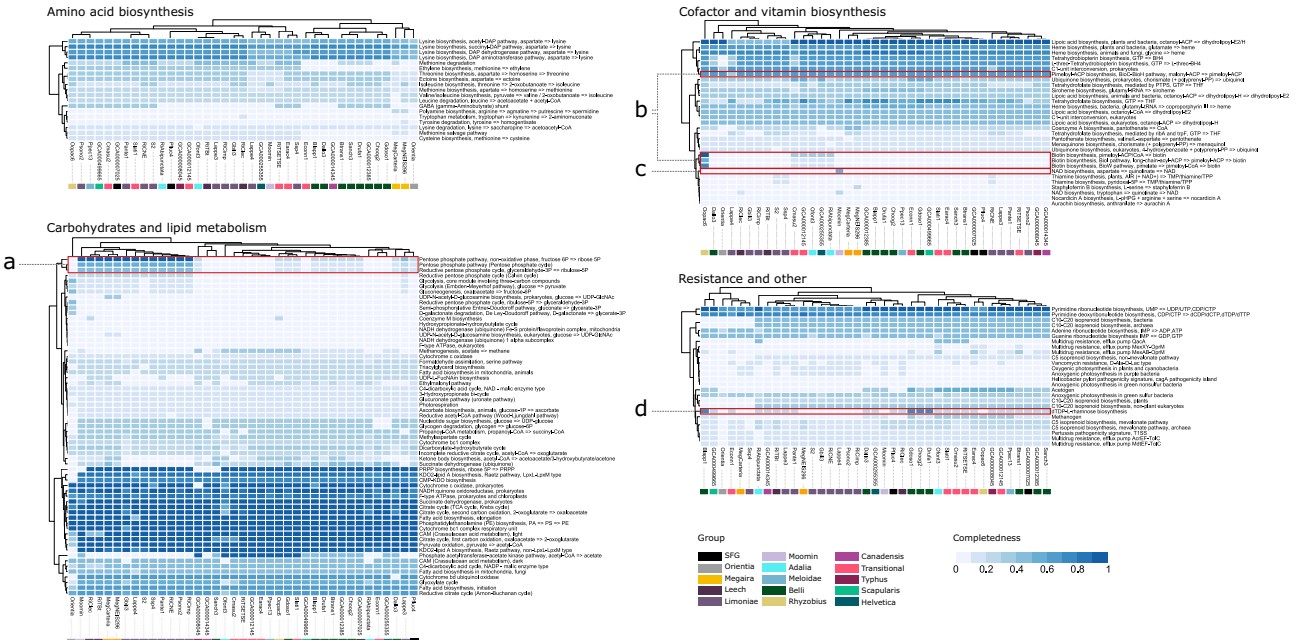

**Fig. 5 Comparison of metabolic potential across selected *Rickettsia* and '*Ca.* Megaira'.** Heatmaps of predicted KEGG pathway completion estimated in Anvi'o 7, separated by function, and produced with Pheatmap. High to low completeness is coloured dark to light blue. Species groups are indicated with a unique colour as shown in the legend. Pathways of interest are highlighted in red: **a** The pentose phosphate pathway only present in Torix and '*Ca.* Megaira', **b** The biotin pathway present only in the Rhyzobius *Rickettsia* Oopac6. **c** NAD biosynthesis only present in Moomin *Rickettsia*. **d** dTDP-L-rhamnose biosynthesis pathway in Gdoso1, Choog2, Drufa1, and Blapp1. SFG is Spotted Fever Group. Source data are provided in Source Data.

has been previously reported as essential for successful nodule invasion of the nitrogen-fixing endosymbiont *Rhizobium*[56]. When we consider both Torix and '*Ca.* Megaira' clades the genes involved in the non-oxidative phase of the PPP pathway were the most highly enriched genes (Supplementary Data 1). It is noteworthy that a large fraction of the enriched genes in both *Rickettsia* and Torix clades are related to cell wall and membrane biogenesis. These are likely associated with differences in the biology of the two clades at the host-microbe interface.

*Rickettsia* lineages group together based on gene presence/absence and produce repeated patterns of accessory genes that reliably occur within each clade (Supplementary Fig. 6). AAI scores separate Torix group, *Rickettsia* and '*Ca.* Megaira' out into genus groups with no score above 65% similarity outside of each respective clade (Fig. 2)[57]. ANI scores suggest that Torix and the remaining *Rickettsia* clades are multispecies clusters with less than 95% similarity between genomes in the same groups except for the Spotted Fever Group (Fig. 2)[57].

Rickettsial genomes extracted from SRA samples are generally congruent with the metabolic potential of their respective groups (Fig. 5). Torix and '*Ca.* Megaira' all have complete pentose phosphate pathways (PPP); a unique marker for these groups which seems to have been lost in the other *Rickettsia* clades[45]. The PPP generates NADPH, precursors to amino acids, and is known to protect against oxidative injury in some bacteria[58], as well as conversion of hexose monosaccharides into pentose used in nucleic acid and exopolysaccharide synthesis. The PPP has also been associated with establishing symbiosis between the *Alpha-proteobacteria Sinorhizobium meliloti* and its plant host *Medicago sativa*[59]. This pathway has previously been highlighted in Torix[45] and its presence in all newly assembled Torix and '*Ca.* Megaira' draft genomes consolidates its importance as an identifying feature for these groups (Fig. 5, Supplementary Data 1). Considering the trend towards gene loss, the PPP is likely an ancestral feature that was lost in the main *Rickettsia* clade[45,60].

Metabolic pathways for Glycolysis, gluconeogenesis, and cofactor/vitamin synthesis are absent or incomplete across all *Rickettsia* included in these analyses, except in the Rhyzobius group member, Oopac6. Oopac6 has a putatively complete biotin synthesis pathway (Fig. 5, Supplementary Fig. 7) and is likely a separate genus according to GTDBtk analysis (Supplementary Data 1). The Oopac6 biotin synthesis pathway is related to, but distinct from, the *Rickettsia* biotin pathway from *Rickettsia buchneri*[36] with which it shares between 85% to 92% amino acid sequence similarity across genes (Supplementary Fig. 7)[36]. Moreover, there is no sequence similarity outside of the biotin operon. This, along with the presence on a plasmid in *Rickettsia* buchneri makes it likely that Oopac6 operon is a result of horizontal gene transfer. Animals cannot synthesize B-vitamins, so they either acquire them from diet or from microorganisms that can synthesize them. Oopac6 has retained or acquired a complete biotin operon where this operon is absent in other members of the genus. Biotin pathways in insect symbionts can be an indicator of nutritional symbioses[61], so Rhyzobius *Rickettsia* could contribute to the feeding ecology of the beetle *O. opaca*. However, like other aleocharine rove beetles, *O. opaca* is likely predaceous, omnivorous or fungivorous (analysis of gut contents from a related species, *O. grandipennis*, revealed a high prevalence of yeasts[62]). We posit no obvious reason for how these beetles benefit from harbouring a biotin-producing symbiont. One theory is that this operon could be a hangover from a relatively recent host shift event and may have been functionally important in the original host. Similarly, if the symbiont is undergoing genome degradation, a once useful biotin pathway may be present but not functional[63,64]. Although the pimeloyl-ACP biosynthesis pathway is partially present (Fig. 5), a *bioH* homolog is not found within or outside the biotin operon (Supplementary Fig. 7) suggesting that this pathway may not be functional (as observed in some *Buchnera aphidicola*[64,65]) or that it may be used in a different way. As this is the only member of this group with a

whole genome so far, further research is required to firmly establish the presence and function of this pathway.

A 75% complete dTDP-L-rhamnose biosynthesis pathway was observed in 4 of the draft belli assemblies (Gdoso1, Choog2, Drufa1, Blapp1) (Fig. 5). Two host species are bird lice (*Columbicola hoogstraali*, *Degeeriella rufa*), one is a butterfly (*Graphium doson*), and one is a ground beetle (*Bembidion lapponicum*). dTDP-L-rhamnose is an essential component of human pathogenic bacteria like *Pseudomonas*, *Streptococcus* and *Enterococcus*, where it is used in cell wall construction[66]. This pathway[67] may be involved in the moulting process of *Caenorhabditis elegans*[68], and it is a precursor to rhamnolipids that are used in quorum sensing[69]. In the root symbiont *Azospirillium*, disruption of this pathway alters root colonisation, lipopolysaccharide structure and exopolysaccharide production[70]. No *Rickettsia* from typically pathogenic groups assessed in Fig. 5 has this pathway, and the hosts of these four bacteria are not involved with human or mammalian disease. Presence in feather lice provides little opportunity for this *Rickettsia* to be pathogenic to their vertebrate hosts because feather lice are not blood feeders, and Belli group *Rickettsia* are rarely pathogenic. Further, this association does not explain its presence in a butterfly and ground beetle; it is most likely that this pathway, if functional, would be involved in establishing infection in the insect host or host-symbiont recognition.

A partial NAD biosynthesis pathway is present only in the Moomin genome. NAD is used as a coenzyme in numerous reactions as well as a substrate in some synthesis pathways, such as ADP-Ribosyltransferases which are used in bacterial toxin-antitoxin systems[71,72]. NAD pathways have previously found in two other members of *Rickettsiaceae*, '*Ca*. Sarmatiella mevalonica' and *Occidentia massiliensis*[31,73]. The most likely explanation for rare occurrence in *Rickettsiaceae* is either a lateral transfer event or remnants from ancestral occurrence.

**Designation of '*Candidatus* Tisiphia'.** In all analyses, Torix group consistently clusters away from the rest of *Rickettsia* as a sister taxon. Despite the relatively small number of Torix genomes, its within group diversity is greater than any divergence between previously described *Rickettsia* in any other group (Fig. 1, Supplementary Figs. 3 and 4). Additionally, Torix shares characteristics with both '*Ca*. Megaira' and *Rickettsia*, but with many of its own unique features (Figs. 3 and 5). The distance of Torix from other *Rickettsia* and '*Ca*. Megaira' is confirmed in both the phylogenomic and metabolic function analyses to the extent that Torix should be separated from *Rickettsia* and assigned its own genus. This is supported by GTDB-Tk analysis which places all Torix genomes separate from *Rickettsia* (Supplementary Data 1) alongside AAI percentage similarity scores less than 65% in all cases (Fig. 2a). To this end, we propose the name '*Candidatus* Tisiphia'. This name follows the fury Tisiphone, reflecting the genus '*Ca*. Megaira' being named after her sister Megaera.

## Conclusions

The bioinformatics approach has successfully extracted a substantial number of *Rickettsia* and '*Ca*. Megaira' genomes from existing SRA data, including genomes for putative Rhyzobius *Rickettsia* and several '*Ca*. Tisiphia' (formerly Torix group *Rickettsia*). Successful completion of two '*Ca*. Megaira' and two '*Ca*. Tisiphia' genomes provide solid reference points for the evolution of *Rickettsia* and its related groups. From this, we can confirm the presence of a complete Pentose Phosphate Pathway in '*Ca*. Tisiphia' and '*Ca*. Megaira', suggesting that this pathway was lost during *Rickettsia* evolution. We also describe previously unsequenced Meloidae and Rhyzobius *Rickettsia* and show that Rhyzobius group *Rickettsia* has the potential to be a nutritional symbiont due to the presence of a complete biotin pathway. These genomes provide a much-needed expansion of available data for symbiotic *Rickettsia* clades and clarification on the evolution of *Rickettsia* from '*Ca*. Megaira' and '*Ca*. Tisiphia'.

## Methods

**Genomic data collection and construction.** We employed two different workflows to assemble genomes for '*Ca*. Megaira' and *Rickettsia* symbionts (Fig. 6). a) Targeted sequencing and assembly of focal '*Ca*. Megaira' and Torix *Rickettsia*.

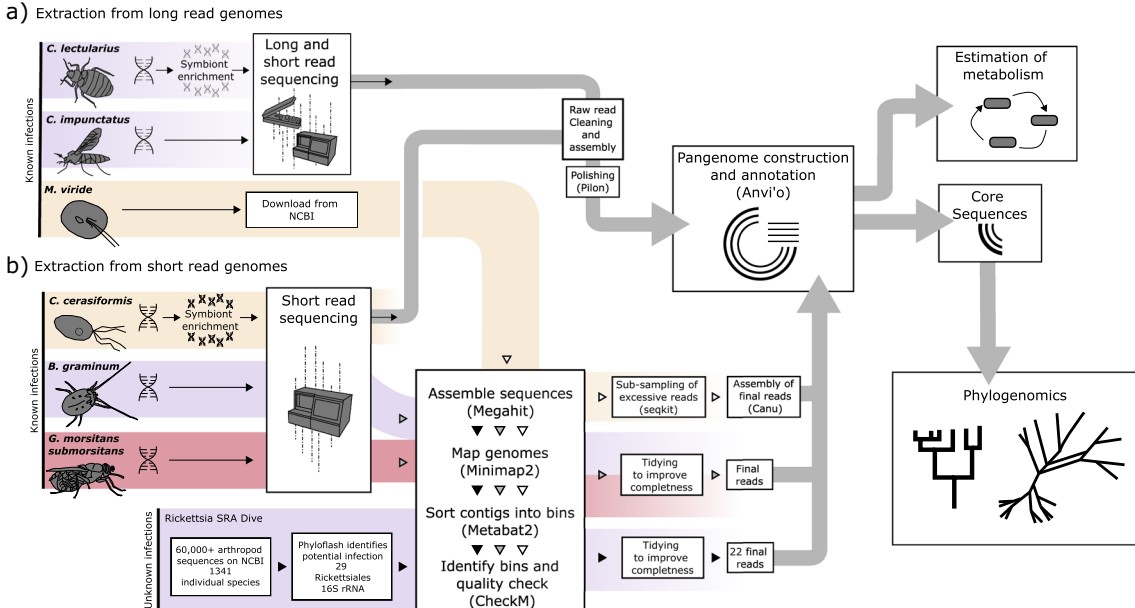

**Fig. 6 Workflow diagram for extraction, assembly and analyses performed in this study.** Workflows for genome assemble are illustrated for (**a**) long read host insect sequences and (**b**) short read host insect sequences. Purple highlights Torix *Rickettsia* and orange highlights '*Ca*. Megaira' and red highlights Transitional *Rickettsia*. Sequencing technologies used vary with source and include Illumina short read sequencing, BGI DNBseq, Oxford Nanopore and PacBio.

b) Assembly from SRA deposits of 'Ca. Megaira' from *Mesostigma viride* NIES296 and the 29 arthropods identified in Pilgrim et al.[13] that potentially harbour *Rickettsia*. These were analysed alongside previously assembled genomes from the genus *Rickettsia*, and the outgroup taxon *Orientia tsutsugamushi*, a distant relative of *Rickettsia* species[74].

DNA preparation, sequencing strategies and symbiont assembly methodologies varied between species and are listed below. The pipeline used to assemble genomes from Short Read Archive (SRA) data is deposited on Zenodo[75].

**Sample collection for targeted genome assembly.** *Cimex lectularius* were acquired from the 'S1' isofemale colony maintained at the University of Bayreuth described in Thongprem et al.[26]. *Culicoides impunctatus* females were collected from a wild population in Kinlochleven, Scotland (56° 42' 50.7"N 4° 57' 34.9"W) on the evenings of the 2nd and 3rd September 2020 by aspiration. *Carteria cerasiformis* strain NIES 425 was obtained from the Microbial Culture Collection at the National Institute for Environmental Studies, Japan. The *Glossinia morsitans submorsitans* specimen Gms8 was collected in Burkina Faso in 2010 and *Rickettsia* infection was present alongside other symbionts as described in Doudoumis et al.[76]. The assembly itself is a result of later thesis work[77].

A *Bryobia* mite community was sampled from herbaceous vegetation in Turku, Finland. The Moomin isofemale line was established by isolating a single adult female and was maintained on detached leaves of *Phaseolus vulgaris* L. cv Speedy at 25 °C, 60% RH, and a 16:8 light:dark photoperiod. The Moomin spider mite line was morphologically identified as Bryobia graminum by Prof Eddie A. Ueckermann (North-West University).

**Previously published *Rickettsia* genomes.** A total of 86 published *Rickettsia* genomes, and one genome from *Orientia tsutsugamushi* were retrieved from the European Nucleotide Archive and assessed with CheckM v1.0.13[78]. Inclusion criteria for genomes were high completeness (CheckM > 90%), low contamination (CheckM < 2%) and low strain heterogeneity (Check M < 50%) except in the case of Adalia for which there is only one genome (87.6% completeness). Filtering identified 76 high quality *Rickettsia* genomes that were used in all subsequent analyses (Supplementary Data 1).

**High molecular weight DNA extraction, assembly, and annotation of complete genomes for two 'Ca. Tisiphia' (==Torix group *Rickettsia*) from *Culicoides impunctatus* and *Cimex lectularius*.** High molecular weight (HMW) genomic DNA was prepared using four-hundred and eighty whole *C. impunctatus* and 45 *C. lectularius* heads, the latter of which had been symbiont-enriched using a protocol designed to eliminate host nuclei through filtration[79]. *Culicoides impunctatus* individuals were pooled and homogenised in two 1.5 ml Eppendorf tubes containing 0.9 ml of buffer G2 (Qiagen) using a pestle while the filtrate from the enriched *C. lectularius* heads was also split and diluted to the same volumes. Twenty-five μL of proteinase K (50 mg/ml) was added to each Eppendorf before incubation at 56 °C for 90 minutes with gentle inversion every 30 min. The respective lysates were centrifuged at 12,000 x g for 20 minutes before the supernatants were pooled and diluted to 3 ml with buffer G2. After equilibrating a Genomic-tip 20/G (Qiagen) with 1 ml QBT buffer, the lysate was gently inverted before being poured onto the tip membrane. The tip was washed four times with 1 ml of QC buffer (Qiagen) before elution of the DNA using buffer QF (Qiagen). Using wide-bore pipette tips, 667 μL of the eluate was pipetted into three 1.5 mL Eppendorf tubes before the addition of 467 μL isopropanol to each tube and mixing by gentle inversion 10 times. Genomic DNA was pelleted by centrifuging for 20 min at 15,000 x g at 4 °C and washing twice with 70% ethanol before resuspending in buffer EB (Qiagen). Quality control of HMW DNA was then confirmed by running on a gel and assessment by Qubit fluorometric quantitation.

Long-read libraries for Oxford Nanopore sequencing were generated using the SQK-LSK109 Ligation Sequencing Kit and sequenced on Minion R9.4.1 flow cells at the Centre for Genomic Research, University of Liverpool, United Kingdom. Raw Nanopore reads were base called using Guppy version 4.0.15 (Oxford Nanopore) using the high accuracy model option (-c dna_r9.4.1_450bps_hac.cfg). All reads which were over 500 bp in length and had an average phred (Q) score of > 10 were filtered using NanoFilt version 2.7.1[80]. These reads were assembled with Flye version 2.8.1[81] using default options.

Assembled circular contigs of ~1.5 Mb in length were confirmed for *Rickettsia* identity by BLASTing against a *Rickettsia* genomic database[45]. High quality short-read libraries were also generated from the same DNA samples and used to correct the nanopore assemblies. *C. impunctatus* paired-end library (2 x 150 bp) was prepared using a Kapa HyperPrep kit (Roche) and sequenced by BGI Genomics (Hong Kong) on a DNBseq platform, whereas *C. lectularius* sequencing was carried out by BGI Genomics (Hong Kong) on a Hiseq Xten PE150 platform. Data cleaning and filtering was performed by BGI Genomics' using SOAPnuke version 2.1.4[82] removing adapters and any reads with 50% of bases having phred scores lower than 20.

Remaining reads were assembled with MEGAHIT version 1.2.9[83] using default parameters and contigs were binned using MetaBAT 2 version 2.12.1[84]. The identities of bins were checked with CheckM version 1.1.3[78] and DNBseq reads were mapped to contigs from the *Rickettsia* allocated bin using 'perfect mode' in

BBMap version 38.87[85] and filtered using SAMtools version 1.11[86]. Filtered *Rickettsia* reads were then used to polish the Flye assembled *Rickettsia* genomes using two rounds of polishing with Pilon version 1.23[87] and the '—bases' option for correcting SNPs and small indels. Annotation of the polished genomes was accomplished using PROKKA version 1.13[88] and identification of polyketide and non-ribosomal peptide synthases was conducted by antiSMASH version 6.0[89].

**Extraction and assembly of a complete 'Ca. Megaira' from *Mesostigma viride*.** 'Ca. Megaira' genome was extracted from recently published reads of *Mesostigma viride* NIES296 (from accession PRJNA509752). Illumina reads were de novo assembled using MEGAHIT version v1.2.9[83], reads were mapped back to the assembled contigs. Contigs were clustered and binned based on nucleotide composition and coverage using MetaBAT2 v2:2.15[84] and a minimum contig length of 1.5 kb. The quality of 'Ca. Megaira' genome bin was inspected using CheckM[78]. The PacBio reads were mapped on the Illumina draft assembly and reads of 'Ca. Megaira' origin were extracted. Due to the excessive number of obtained reads a sub-sample (reads > 10 k and 1/3 of the total) was taken using seqkit[90] and used for subsequent analysis. This sub-sample of PacBio reads was assembled using Canu version 1.8[91] under default parameters. The final assembly, consisted of two contigs, was manually inspected for circularization and trimmed accordingly. The final and circular assembly was further polished by a combination of PacBio and Illumina reads using Pilon v1.22[87].

**Extraction of Transitional *Rickettsia*, RiTSETSE, from *Glossina morsitans submorsitans*.** All methods described here for the extraction of *G. morsitans submorsitans* originate from a thesis by Frances Blow[77].

DNA was extracted immediately using the CTAB (Cetyl trimethylammonium bromide) method and was stored at −20 °C. Whole Genome Shotgun (WGS) libraries were prepared with the Illumina TruSeq Nano DNA kit following the manufacturers' instructions. Samples were sequenced on two lanes of Illumina HiSeq with 250 bp paired-end reads. Raw sequencing reads were de-multiplexed and converted to FASTQ format with CASAVA version 1.8 (Illumina). Cutadapt version 1.2.1[92] was used to trim Illumina adapter sequences from FASTQ files. Reads were trimmed if 3 bp or more of the 3' end of a read matched the adapter sequence. Sickle version 1.200[93] was used to trim reads based on quality: any reads with a window quality score of less than 20, or which were less than 10 bp long after trimming, were discarded.

Metagenomic reads were assembled with DISCOVAR[94] and contigs shorter than 500 bp were removed and mapping with Bowtie2[95] was used to assess coverage. Taxonomy was assigned to contigs with BLAST and the GC content of contigs assessed with the Blobology package[96]. Contigs were filtered based on GC content, coverage and taxonomy, and reads were extracted using scripts implemented in Blobology. Extracted reads were re-assembled with SPAdes version 3.7.1[97] and mapped to contigs with Bowtie2. Assembly statistics were calculated with custom perl scripts and Qualimap version 2.2[98].

**DNA extraction of Moomin 'Ca. Tisiphia' (== Torix group *Rickettsia*) from *Bryobia graminum* str. moomin.** Genomic DNA was extracted from ~1000 adult females using the Quick-DNA Universal kit (BaseClear, the Netherlands) and was sequenced by GENEWIZ on an Illumina NovaSeq instrument. *Rickettsia* sequence was extracted from illumina reads as described for other MAGs.

**DNA extraction of 'Ca. Megaira' from *Carteria cerasiformis*.** Symbiont enriched DNA was extracted from culture using a modified version of the protocol of Stouthamer et al.[79]. Specifically, prior to homogenization the *Carteria cerasiformis* culture was filtered through a 100um filter/mesh to reduce bacterial contamination. DNA extraction was performed using the QIAGEN DNAeasy™ Blood & Tissue Kit. Short read sequencing was carried out by BGI Genomics (Hong Kong) on a Hiseq Xten PE150 platform. *Rickettsia* sequences were assembled from Illumina reads as described for other MAGs.

**Assembly, and annotation of *Rickettsia* genomes from publicly available SRA data.** Pilgrim et al.[13] identified 29 SRA deposits containing *Rickettsia* DNA. We used these datasets to extract and assemble 22 new high quality draft *Rickettsia* genomes. Briefly, short reads from each SRA library were assembled using MEGAHIT v1.2.9[83], mapped with Minimap 2 v2.17-r941[99] and contigs were binned based on tetranucleotide frequencies using MetaBAT2 v2:2.15[84]. *Rickettsia* like bins were quality inspected with CheckM v1.0.13[78]. Bins with a completeness score of over 50% and contamination below 2% marked as *Rickettsiales* or *Rickettsia* were then retained onward for further refinement, annotation, and scrutiny.

To refine MAGs, insect SRA contigs were compared against a local *Rickettsia* genome database using Blastn[100]. Contigs with significant matches to the database were extracted, non-*Rickettsia* contigs were identified with blastx against the nr database and contigs with atypical coverage were discarded. MetaBAT 2 filtered out reads less than 1.5 kb long for accuracy, but these reads are potentially informative in small symbiont genomes, so contigs with a length of 1–2.5 kb were manually examined and added to MetaBAT 2 assembled genomes. Those with improved CheckM score and no *Wolbachia* in the original host are used as the final draft genome for the *Rickettsia*. The additional genome for the leech *Rickettsia*, RiTBt,

was found to contain *Cardinium* contamination during separate examination. RiTBt contigs identified as *Cardinium* using blastx were removed from the genome, reducing contamination from 9.48% to 0.95%. The final pipeline resulted in 22 MAGs each with completeness >90% and contamination <2%.

**Genome content comparison and pangenome construction.** Anvi'o 7[101] was used to construct a pangenome. Included in this were the 22 MAGs retrieved from SRA data, 2 'Ca. Megaira' genomes and 4 targeted Torix *Rickettsia* genomes, and one Transitional group *Rickettsia* genome acquired in this study. To these were added the 76 published and 1 *Orientia* described above, giving a total of 104 genomes. Individual Anvi'o genome databases were additionally annotated with HMMER, KofamKOALA, and NCBI COG profiles[102–104]. For the pangenome itself, orthologs were identified with NCBI blast, mcl inflation was set to 2, and minbit to 0.5. Average nucleotide sequence identity was calculated using pyANI[105] within Anvi'o 7 and Average Amino Acid identity was calculated though the Kostas Lab enveomics online calculator[106]. Networks of ANI and AAI results were produced in Gephi 0.9.2[107] with Frutcherman Reingold layout and annotated in Inkscape 0.92[108]. Exact code and a list of packages used is available on Zenodo[75].

KofamKOALA annotation[103] in Anvi-o 7 was used to estimate completeness of metabolic pathways and Pheatmap[109] in R 3.4.4[110] was then used to produce heatmaps of metabolic potential. Annotations for function and *Rickettsia* group were added post hoc in Inkscape.

The biotin operon found in the genome Rhyzobius *Rickettsia*, Oopac6, was identified from metabolic prediction. To confirm Oopac6 carries a complete biotin pathway that shares ancestry with the existing *Rickettsia* biotin operon, Oopac6 biotin was compared to biotin pathways from five other related symbionts: *Cardinium*, *Lawsonia*, *Buchnera aphidicola*, *Rickettsia buchneri*, and *Wolbachia*. Clinker[111] with default options was used to compare and visualise the similarity of genes within the biotin operon region of all 6 bacteria. Clinker by default displays the highest similarity comparisons based on an all-vs-all similarity matrix.

All generated draft and complete reference genomes were annotated using the NCBI's Prokaryotic Genome Annotation Pipeline (PGAP)[112]. Secondary metabolite biosynthetic gene clusters were identified using AntiSMASH version 6.0[89] along with Norine[113] which searched for similarities to predicted non-ribosomal peptides. BLASTp analysis was additionally used to identify the closest homologues of these biosynthetic gene clusters.

Functional enrichment analyses between the main *Rickettsia* clade and the Torix – 'Ca. Megaira' clades were performed using the Anvi'o program anvi-get-enriched-functions-per-pan-group and the "COG_FUNCTION" as annotation source. A gene cluster presence – absence table was exported using the command "anvi-export-tables". This was used to create an UpSet plot using the R package ComplexUpset[114] to visualize unique and shared gene clusters between different *Rickettsia* groups. A gene cluster was considered unique to a specified *Rickettsia* group when it was present in at least one genome belonging to that group. Gene cluster accumulation curves were performed for the pan-, core- and unique-genomes based on the same presence-absence matrix using a custom-made R script[115]. In each case the cumulative number of gene clusters were computed based on randomly sampled genomes using 100 permutations. The analysis was performed separately for Torix group and the combined remaining *Rickettsia*. Curves were plotted using the ggplot2 R package[116].

All information on extra genomes can be found in Supplementary Data 1, and the code pipeline employed can be found on Zenodo[75].

**Phylogeny, network, and recombination.** The single-copy core of all 104 genomes was identified in Anvi'o 7 and is made up of 74 single-copy gene (SCG) clusters. Protein alignments from SCG were extracted and concatenated using the command "anvi-get-sequences-for-gene-clusters". Maximum likelihood phylogeny was constructed in IQ-TREE v2.1.2[117]. Additionally, 43 ribosomal proteins were identified through Anvi'o 7 to test phylogenomic relationships. These gene clusters were extracted from the pangenome and used for an independent phylogenetic analysis. The best model according to the Bayesian Information Criterion (BIC) was selected with Model Finder Plus (MFP)[118] as implemented in IQ-TREE; this was JTTDCMut+F + R6 for core gene clusters and JTTDCMut+F + R3 for ribosomal proteins. Both models were run with Ultrafast Bootstrapping (1000 UF bootstraps)[119] with *Orientia tsutsugamushi* as the outgroup.

The taxonomic placement of Oopac6, Ppec13 and Dallo3 genomes within the Rhyzobius, Meloidae and Belli groups respectively were confirmed in a smaller phylogenetic analysis, performed as detailed in Pilgrim et al.[13] using reference MLST sequences (gltA, 16 S rRNA, 17 kDa OMP, COI) from other previously identified *Rickettsia* profiles (Source Data). The selected models used in the concatenated partition scheme were as follows: 16 S rRNA: TIM3e + I + G4; 17Kda OMP: GTR + F + I + G4; COI: TPM3u + F + I + G4; gltA: K3Pu + F + I + G4a.

A nearest neighbour network was produced for core gene sets with default settings in Splitstree4 to further assess distances and relationships between *Rickettsia*, 'Ca. Megaira' and Torix clades. All annotation was added post hoc in Inkscape. Furthermore, recombination signals were examined by applying the Pairwise Homoplasy Index (PHI) test to the DNA sequence of each core gene cluster extracted with Anvio-7. DNA sequences were aligned with MUSCLE[120] and PHI scores calculated for each of the 74 core gene cluster with PhiPack[121].

The taxonomic identity for genomes was established with GTDB-Tk[122] to support the designation of taxa through phylogenetic comparison of marker genes against an online reference database.

**Reporting summary.** Further information on research design is available in the Nature Research Reporting Summary linked to this article.

## Data availability

The genomes and raw read sets generated in this study have been deposited in the GenBank database under accession code PRJNA763820. The assemblies produced from previously published third party data have been deposited in the GenBank database under accession code PRJNA767332. The genome content data and data for figures generated in this study are provided in the Source Data and Supplementary Data. Accessions and metadata for pre-existing genomic data are listed in the Supplementary Data 1 file.

## Code availability

All code and bioinformatics pipelines used to extract and construct bacterial genomes from SRA data can be found on Zenodo (https://doi.org/10.5281/zenodo.6396821), and the R script for generating pangenome accumulation curves can be found on GitHub (https://github.com/SioStef/panplots and here 10.5281/zenodo.6408803). The full pangenome Anvi'o database is available on Figshare (https://doi.org/10.6084/m9.figshare.14865576.v3). An interactive html version of Fig. 5 and its associated 'json' file is available on Figshare (https://doi.org/10.6084/m9.figshare.14865567.v5). html of bonzai module information for Supplementary Fig. 2 is available on Figshare (https://doi.org/10.6084/m9.figshare.14865570.v4).

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

## Acknowledgements

Grant supporting these works: NE/L002450/1 NERC ACCE Doctoral Training Programme, HRD. Ghent University 01P03420 BOF post-doctoral fellowship and 1513719 N Research Foundation - Flanders (FWO) Research Grant, NW. Funding for tsetse fly genomics were to ACD IP BBSRC projects BB/J017698/1 and BB/K501773/1 FB, the materials from which were provided by Philippe Solano (Institut de Recherche pour le Développement, Montpellier, France) and Jean-Baptiste Rayaisse Centre International de Recherche-Développement sur l'Élevage en zone Subhumide (CIRDES), Bobo Dioulasso, Burkina Faso. Jean-Baptiste died a few years ago but he was a fantastic person to work with and a great field entomologist. We also wish to thank Dr David Montagnes for teaching skills associated with algal culture.

We wish to thank Dr Débora Pires Paula (Embrapa) for granting permission to use SRA data for sample number SRR5651504, Iridian Genomes for allowing use of their SRA data, and the Microbial Culture Collection at the National Institute for Environmental Studies, Japan for use of the sample Carteria cerasiformis NIES-425.

## Author contributions

Project concept: H.R.D., S.S., Jack Pilgrim, and G.H. Manuscript written by H.R.D., S.S., J.P., and G.H. All authors commented on the manuscript during development and approved the final version. S.R.A. dive and metagenome assembly carried out by H.R.D. with aid from S.S. Assembly of genome from S.R.A., pangenomics and phylogenomics carried out by H.R.D. with advice from S.S., G.H. Metabolic analysis carried out by H.R.D., Jack Pilgrim and S.S. Sequencing and assembly of bacteria from *Cimex lectularius* and *Culicoides impunctatus* genomes by S.S. and Jack Pilgrim. Sequencing and assembly of symbionts from *Carteria* by S.H.B. and S.S, supervised by P.C. and G.H. Sequencing and construction of RiTSETSE conducted by F.B. as part of thesis work supervised by A.D. J. Parker and S.P. collected and sequenced staphylinid genomes that were released

through NCBI by Iridian Genomes. N.W. collected and sequenced the Bryobia Moomin strain and performed preliminary metagenomic analyses.

## Competing interests

The authors declare no competing interests.
