## [Peer Review File · Nature Communications]

Reviewers' Comments:

Reviewer #1:

Remarks to the Author:

The manuscript "Large-scale comparative genomics unravels great genomic diversity across the Rickettsia and Ca. Megaira genera and identifies Torix group as an evolutionarily distinct clade" by Davidson et al. presents a massive analysis of dozens of novel rickettsial genomes that informs on the classification of genus Rickettsia and the evolutionary characteristics of the genomes of these obligate intracellular bacteria. This work comes off the heels of several papers by Dr. Hurst and colleagues analyzing metagenomic datasets (mostly insect genome sequencing projects) and other publicly available data to determine that a basal rickettsial lineage, termed "Torix Group Rickettsiae" is widespread in nature, particularly in non-bloodfeeding arthropods but also in some vectors and potential vectors of human disease. These works, combined with the present study, and critically important to human health because they not only enlighten on the factors leading to the highly pathogenic species (i.e. agents of typhus and spotted fevers) but they place Rickettsiae alongside Wolbachiae in as far as diversity and widespread occurrence in arthropod populations. Thus, as some have argued, approaches tailored for Wolbachiae for biocontrol of arthropod-borne pathogens are applicable to many Rickettsia species that seemingly persist in arthropod populations on par with their rickettsial cousins in the Anaplasmataceae.

There are several major accomplishments of this work that I point out below:

- 1) The first closed genomes of "Candidatus Megaira" (from the algae *Mesostigma viride*) and Torix Group Rickettsiae (from the midge *Culicoides impunctatus* and bed bug *Cimex lectularius*) are provided by the team. Closed genomes from these lineages are critical to understanding the Rickettsia pan genome and the extents of lateral gene transfer and metabolic capacity.
- 2) The team sequenced and constructed draft genomes for "Candidatus Megaira" from another algal host (*Carteria cerasiformis*), a Transitional Group Rickettsiae from tsetse fly (*Glossina morsitans morsitans*), and a Torix Group Rickettsiae from a spider mite (*Bryobia graminum*). These genomes are invaluable for conducting robust phylogenomic analyses as they will critical gaps in the diversity already in hand.
- 3) The team further extracted 22 draft genomes from arthropod genome sequencing projects, including tentative species from Adalia Group Rickettsiae (n = 1), Transitional Group Rickettsiae (n = 4), Spotted Fever Group Rickettsiae (n = 1), Torix Group Rickettsiae (n = 7), Belli Group Rickettsiae (n = 7), Rhyzobius Group Rickettsiae (n = 1) and Meloidae Group Rickettsiae (n = 1). Importantly, for the latter two groups, these are the first genomic datasets constructed to allow robust phylogenomic analysis.
- 4) Finally, the team effectively utilizes this massive genomic dataset of new and existing genomes to characterize the basal lineages previously named Torix Group Rickettsia and identify key attributes in the metabolome and accessory genome that, along with genetic divergence in estimated phylogenies, warrant removing this clade from genus Rickettsia. The team proposes the genus name "Tisiphia" as an immediate sister lineage to the remaining Rickettsia lineages.

These four accomplishments are probably the most important contributions to Rickettsia biology and evolution in the last decade. They stand to enlighten on specific works focusing on Rickettsia pathogenesis since we may now understand the origin and maintenance of describe pathogenicity factors across a much broader and robust evolutionary framework. Furthermore, the completeness of the genomic datasets provides taxonomic resolution to the basal lineages that have been plagued by incompleteness and lack of thorough analyses.

The manuscript is also very well written and a pleasure to read.

I recommend acceptance of this monumental work after a careful revision that considers the following:

General statement: usage of genus names as nouns is incorrect, though sadly commonplace.

Genus names should be modifiers of either a species name (species epithet) or a noun (Wolbachia gene, Rickettsia phylogeny, etc.). "Genus Rickettsia", "Rickettsia species", "Rickettsiae", etc. are all correct. All provisional names should be used as follows: "Candidatus <Non-italicized genus name> <non-italicized species name>", or with "Candidatus" abbreviated to "Ca." but still in italics.

Line 64: italicize Alphaproteobacteria.

Line 71-77: perhaps here or elsewhere, Gillespie et al. (PMID: 25477419) identified a plasmid named pLbAR that carries a toxin-antidote (TA) module purported to distinguish booklouse-associated Rickettsia felis from flea-associated R. felis; the TA module was later shown to carry the domains of Wolbachia CI factors and hypothesized to underpin parthenogenesis induction in booklice, which are all-female when infected with R. felis (PMID: 30060072).

Line 78: Only two years prior Gillespie et al. (PMID: 17342200) analyzed existing genomes, particularly plasmid pRF from R. felis, and concluded that a lineage distinct from Typhus Group Rickettsiae and Spotted Fever Group Rickettsiae should be recognized. This clade, termed Transitional Group Rickettsiae, has since been well separated from the Spotted Fever Group Rickettsiae with the availability for more genomes as well as more sophisticated phylogeny estimation tools. These two factors alone have the potential to lead to continual reorganizations of Rickettsia classification as the authors demonstrate here.

Line 78-100: per my comment above, designations of "Groups" based on a few taxa and a few genes is pretty tenuous. How many of the 13 groups in Weinert et al. 2009 will hold up when more species and genome sequences are unearthed? Is there a negative impact on the field when these "Groups" are proposed but later have to be revised? Also, some of the groups names are really superficial, describing one type of host for a few members that have different hosts (e.g., the "Ixodes Group" comprises R. tamurae and R. colombianensi that infect mostly Amblyomma ticks, and R. helvetica can infect Ixodes ticks but is not within this clade!).

Line 101: this idea has been proposed before (PMID: 25073875 and PMID: 23475938) and may have been touched on by Perlman et al. in their seminal report (PMID: 16901827).

Figure 1: very nice flow chart. Some of the text is difficult to read.

Line 129: having A and B sections here without panels in Figure 1 is a little confusing.

Line 134: the reader is introduced to Orientia tsutsugamushi for the first time here. It might be worth introducing this species in the Introduction...it is relevant that it was once called Rickettsia tsutsugamushi and now at this moment is so far removed relative to all of this new diversity. Some readers could benefit from this information.

Lines 143-156: Are all materials deposited as vouchers? Is the genetic material archived and available?

Line 211: the specific supplementary figure with the ribosomal protein tree is not called out.

The Materials and Methods are very well described and easy to follow, with all appropriate references provided. It is conceivable that all methods followed as described would lead to similar results obtained by the authors.

Line 241: This is an interesting finding. Gillespie et al. identified only ONE gene that is present on all Rickettsia plasmids: an odd DnaA domain-containing protein (PMID: 25477419). Is this present on these plasmids? Are there RAGE genes that tend to be on most of the Rickettsia plasmids? It would be interesting to learn your opinion on the origin of these plasmids and if they provide any links between Tisiphia and Rickettsia species, or exchanges with other microbes.

Line 261: fascinating!

Figure S3: it would be helpful to know the top blastp hits to these interesting proteins. Are they similar to the *R. buchneri* cassettes or do they have a different evolutionary profile? Perhaps a little digging in this regard might enlighten on a possible function? Also, are they syntenic with any other microbes? You could just blastn the entire nucleotide sequence for the cassettes and see it right away (or tblastx if the closest syntenic counterparts are too divergent at the nt level).

Figure S3: these images are pretty but hard to read...you could utilize the space better and make it easier on the reader.

Line 278: word usage suggestion here, "The Transitional Rickettsia" could be written "the TRG Rickettsia species".

Line 278: This is interesting, but I wonder if this information will be associated with the data on NCBI and other databases? Or will others that don't read this work have to suffer the consequences. Or is it that these are so close (strains I assume) that it doesn't really matter for tree-building and such. Are there more than one 16S rDNA sequences?

Line 286: How were Lappe3 and Lappe4 unambiguously assembled?

Table 2: great care was taken to assess the relative completeness of the existing Rickettsia genome assemblies; how well are these new assemblies in relation to the existing ones? How often does each new assembly disrupt a core gene set? Is there some metric that can be used to assess and rank the relative completeness of these assemblies?

Line 307: The original paper on the seal fur louse Rickettsia species should probably be cited here.

Line 310: You use "Transitional" as if it is an adjective but follow with "Spotted Fever Group". It should be TRG or Transitional Group Rickettsiae.

Line 316: Guilotte et al. recently showed *R. helvetica* as basal to TRG, TG, and SFG Rickettsiae (Figure 2 in PMID: 33952661). This tree was modified from Hagen et al. that also reported the same phylogenetic position for *R. helvetica* (PMID: 30398619).

Line 316: The text here is confusing...why would you posit *R. helvetica* is most similar to the "Scapularis group" if it belongs in a unique clade well removed from this group? Furthermore, including only one member of the "Scapularis group" in your phylogeny estimates makes it seem as if this clade is not stable. There are plenty of conserved genes for the *I. pacificus* and *I. scapularis* endosymbionts to provide stability and show that *R. helvetica* does not belong to this clade. Do you have some gene profile support for your supposition?

Figures 2 and S4: Two problems here. One, the divergence can hardly be seen for most clades. This could be solved by collapsing all monophyletic strains (e.g., *R. prowazekii* and *R. rickettsii*) and truncating the species names (i.e. *R.* instead of Rickettsia) so the figure can be expanded. Two, a simple cladogram can be shown to the right of clades with little divergence. Otherwise, these trees will remain difficult to read.

Figures 2 and S4: It would cool and perhaps informative to map the clades that differ across estimations. Does the discordance jive with low support values?

General comment: what are your criteria for naming groups? Monophyly? A certain degree of divergence on estimated phylogenies? Can a group be less than two entities? Is "Canadensis" a group if only two strains form the clade?

Figure 3: I am not really sure of the value of this analysis in light of the phylogeny estimation; so few genes are analyzed and they should be stable and conserved. Obviously, such a network will get messy when less conserved genes are analyzed.

Line 333: What is the history of "Rhizobius Group" and why is it important to keep this naming system? Could the "Meloidae Group" possibly be combined with the Bellii Group and given one

name? Or is it too divergent for that? It seems that the *Onychiurus sinensis* associated *Rickettsia* species may indicate further group resolution down the road.

Figure S5: this tree is difficult to read like the others (can some of the close divergences be better illustrated?). It seems also that the trees are not ordered...at first glimpse it looks confusing and contradictory to the trees in Figures 2 and S4. I think the only glaring difference (monophyletic TG + SFG) would emerge better with ordering and showing the divergences better.

Line 354: There are a lot of others, most reviewed in PMID: 23475938, which also provided scaffold and transcriptional evidence for rickettsial genes in the *Trichoplax adhaerens* genome. More recently, a *Wolbachia* CI antidote was shown inserted as an exon in a larger cat flea gene (PMID: 33362982); the CI genes themselves are often found in eukaryotic genomes (PMID: 30060072). Not all of these are *Wolbachia*-like...some are *Rickettsia*- and *Cardinium*-like!

Line 371: revise English for clarity.

Line 379: what features distinguish this accessory genome?

Figure 4: this is difficult to read...there seems to be room to enlarge the taxa at the top right. Also, some metric would be nice to associate with the sizes of the accessory genomes per group (averages?). The arrangement of the rings seems strange...why are the groups out of phylogenetic order (radiating from center)?

Figure 5: this could be a supplement to save space; it is sort of implied from the phylogeny estimations and is in agreement.

Figure 6: it would seem more useful to me if the taxa were arranged phylogenetically at bottom rather than by cluster size. It is also difficult to read a lot of the text in this figure. It seems better arrangement and space minimization could make things clearer, larger font sizes too.

Figure 7: this is a very cool figure and I am glad to see the permutations conducted. Well done!

Line 427: This was concluded by Driscoll et al. (PMID: 28951473) and can be inferred from comparisons with sister *Rickettsiales* lineages. It also can be explained by the presence and ability of *Rickettsiaceae* to import ribonucleotides required for interconversions to deoxyribonucleotides.

Line 433: I have tried hard to understand this sentence: "Based on the gene cluster comparison plot and an independent blastx search, the GlyA gene in *Rickettsia buchneri* appears to be a misidentified bioF gene". Is this some different annotation that was reported by Gillespie et al. a decade ago? Please clarify. NCBI likes to turn wine to water when it comes to annotations.

Line 435: I don't understand this sentence either: "Additionally, the insect SRA sample was not infected with *Wolbachia*, making it unlikely that the presence of the biotin operon is a result of misassembly". Why bring up *Wolbachia* here? The figure shows greatest similarity between Oopac6 and *R. buchneri* BOOM, so what does *Wolbachia* contamination have to do with anything?

Line 438: "Oopac6 has retained or acquired a complete biotin operon where this operon is absent in other members of the genus"; you need to estimate a phylogeny like Driscoll et al. recently did (Figure 2C in PMID: 33362982) and determine this. Based on the similarity between Oopac6 and *R. buchneri*, it is likely the BOOM invaded *Rickettsia* species multiple times. The loss of bioH in Oopac6 is telling; have you looked for it elsewhere in the genome or identified any other non-orthogonal methyl esterases? There are several different kinds that bacteria use to regulate biotin synthesis (see Figure S4 in PMID: 33362982).

Line 445: The synteny of the BOOM is telling in relation to other biotin synthesis gene operons and clusters. It would seem strange that the synteny would be similar to BOOM if Oopac6 had biotin synthesis capability and secondarily lost it. There is no evidence for an alternative bio gene arrangement for comparison. It seems more like that Oopac6 picked it up and maybe is in the process of losing it (loss of bioH); it could also be a symbiont that is experiencing a recent host shift

and no longer benefits from supplying biotin to a host that gets plenty of it from its diet.

Figure 8: very difficult to read. Some order of the metabolic processes would help as well.

Line 467: This is interesting. It could mean that loss of rhamnose in the O-antigen is more of a characteristic of hematophagous host-associated species.

Line 496: Is the pathway complete in the absence of bioH or another methyl esterase?

Line 497: Agreed on all counts. This is a fabulous contribution to Rickettsiology and will have a tremendous and lasting impact. A massive effort. Kudos to the authors.

Reviewer #2:

Remarks to the Author:

In the manuscript 'Large-scale comparative genomics unravels great genomic diversity across the Rickettsia and Ca. Megaira genera and identifies Torix group as an evolutionarily distinct clade' Davison and colleagues present novel Rickettsiaceae genomes, both from specific sequencing efforts and database mining. The genomes, that highly enrich the known genomic diversity of Rickettsiaceae are analyzed through phylogenetics and comparative genomics methods. The authors convincingly conclude that the Torix group is different enough to merit genus status. The work is of interest and provides a significant advance in the field.

The methods used are generally solid and well explained. The sequencing is performed with both long and short reads (on some samples) allowing to get complete genomes. Comparative genomics is performed with multiple tools allowing to compare synteny, gene content, nucleotide identity and 'functional' annotation.

The manus is well written and clear, with very few minor typos (see below). The quality of the supplementary materials text appears a bit lower, and could use some polishing.

The main figures are in general clear and vehiculate the message, see below for specific comment. The supplementary figures and materials are informative.

The work, however, presents one main issue (or multiple connected issues).

Considering that the authors delineate Torix as a new genus (and I agree with this) and show the presence of subgroups in Torix, I would find appropriate and interesting to compare the diversity of classical rickettsia groups to the Torix groups, not to Torix as a whole. As they stand now, the comparative analyses have a little bit of an 'apple to oranges' problem. Either compare true Rickettsia to Tisiphia as a whole, genus to genus, or Rickettsia groups to Tisiphia groups (as let's say species to species). The Ani of the 'true rickettsia' groups is much higher than the novel ones, clearly highlighting this issue.

Related to this is the pangenome analysis. First I feel it makes no sense to evaluate the pangenome of the entire dataset (which includes Megaira and Orientia if I correctly understand). The genomes are too diverse because the sampling is too broad, being almost the pangenome of the entire Rickettsiaceae family. Secondly, we have again the 'apple to oranges' problem. Of course the pangenome of Tisiphia is wider than that of Rickettsia group, because it is that of an entire genus. While I understand this fortifies the notion that Tisiphia is a genus, then it leads to the very strong conclusion that the pangenome is open. An open pangenome implies a higher genomic plasticity, which would be a huge finding for a group of intracellular symbionts, but is this the case? Is it a true result or is it due to the higher diversity of the novel groups?

If the analyses were performed lumping all the true Rickettsia together, would the authors find an open pangenome? If this was the case, would it be a true biological result? I would argue it would not. Pangenomes should be of species, not of genera. I suggest to either strongly modify this analysis or to remove it completely.

Related to this, is the issue of Megaira diversity. Is it really lower than Torix diversity (as it is somewhat implied for example at line 377), or is it only perceived so due to the lack of genomic

resources? Previous trees of *Megaira* diversity (16S based) would suggest the latter. The authors should address this in the text.

Additional issues are highlighted below.

A comment/suggestion on the tone. The manuscript contains the word 'first' 19 times. I did not check them all, but a good number of them are used to indicate the novelty of the results. I would reduce this number a bit. While the novel genomes are the foundations of this work, I would try to move the emphasis balance more towards how they were analyzed and what the authors learnt from them. Saying 'this is the first' a couple of times less would not make the results any less novel but would make the manus sound, in my humble opinion, a little bit classier.

Gene content analysis: figure 8 shows four main findings. Three are discussed in too much detail in the text, one not enough. Please reduce this part, and make it more balanced, adding a sentence on Nad synthesis (e.g. present in doi: 10.1186/1944-3277-9-9 and doi: 10.1111/1462-2920.15396).

Line 84, 106...: *Megaira* is not the sister group of *Rickettsia*. The authors completely ignore *Trichorickettsia* and *Gigarickettsia*. No genomes are available for these two genera, but their existence should be at least acknowledged.

Line 246: I find this result to be really interesting, especially for how extreme the lack of sytheny is (Fig S3). A short discussion on this result would be nice, for example comparing it to previous work by one of the authors on *Wolbachia* (doi:10.1098/rsob.150099.) or discussing how common this lack of sytheny could be in *Rickettsiales*. Maybe the generally accepted notion of symbiont genomes plasticity to be all about losing genes, and the low number of complete *Rickettsiales* genomes, have led to underestimating this trend until now?

This could be discussed in connection with the pangenome results (if updated).

The main tree (fig 2).

Iqtree bootstraps should be color-coded differently, dividing 91-100% and 81-90%, as in IQTREE these values are considered differently (>90% equals high support).

Thypus and spotted fever are highly overrepresented.

Could this lead to errors in the phylogenetic reconstructions? Unlikely, but worth checking.

I would like to see the same analyses with a smaller, more balanced dataset (e.g remove multiples from same species).

The trees from the complete datasets could be moved to the supplementary materials, and the smaller tree would make better main figure, with a stronger focus on the novel groups.

Also the tree should be expanded horizontally, to allow to better gauge branch lengths.

Figure 4, while beautiful, is not very informative. Since there are a total of 8 main figures I would move this to supplementary.

Could *Moomin* and *Rhizobius* be different genera? I guess the authors think not (and I feel I agree), but based on trees and ANI, the authors should at least consider the possibility, and explain why they think they do not.

MINOR COMMENTS

Line 64 and below: *Rickettsiales* and *Alphaproteobacteria* should be in italics. So *Rickettsiaceae* etc. *Candidatus* should be in quotes.

Line 64: there is one non-intracellular *Rickettsiales* (doi: 10.1038/s41396-019-0433-9)

Line 74: 'with the symbiont capable' clarify to something like 'with the different symbionts being

capable...'

Line 102: 'infect invertebrate symbionts' remove symbionts

Line 211: please provide number for the SUPPLEMENTARY FIG.

Line 307: add citations

Line 346: correct multilocus

Reviewer #3:

Remarks to the Author:

Davison et al greatly increase the sample of Rickettsia-related genomes by directly sequencing HMW DNA extracted from a range of arthropods and an alga, re-assembling data from a previously published genome sequencing effort of the alga *Mesostigma viride* and mining 29 publicly available arthropod SRA datasets known to contain traces of Rickettsia DNA.

All in all the authors manage to assemble 3 complete genomes and another 25 draft genomes. This awesome effort was particularly successful in reconstructing genomes from previously underrepresented Rickettsia clades. These include first genomes of the Meloidae, Rhyzobius and Megaira clades, and many additional genomes of the Limoniae, Leech, Moomin (collectively "Torix Rickettsia") and Belli groups.

In-depth comparative and phylogenetic analyses reveal the Torix clade to harbor a relatively large divergence and to be a sister clade of the Rickettsia genus. The authors propose a new name, *Ca. Tisiphia* for this group, a novel genus. *Ca. Tisiphia* and *Ca. Megaira* genomes were found to encode an intact pentose phosphate pathway, a feature lost in all other Rickettsia genomes. "Oopac6", the first genome of the Rhyzobius group, was found to encode a complete biotin synthesis pathway that is otherwise absent in the Rickettsia genus. Finally, four novel Belli group genomes were found to encode a near complete dTDP-L-rhamnose biosynthesis pathway, a unique feature among all Rickettsia genomes considered in this study.

I'm very impressed with the amount and the quality of the work, and the manuscript is generally well and clearly written. It is a substantial step forward into the field. I have learned a lot! I have no major comments. I do however, have several minor comments. Approximately from most-to-least important:

374-381, Figure 6 & 7: Here the Torix Rickettsia are compared with other Rickettsia groups in terms of gene content overlap with other groups and pangenome rarefaction curves. The authors conclude that out of all the clades, the Torix group has the most unique genes and has likely the largest pangenome (if I understand line 379 and Figure 7 correctly). Though the biological and/or evolutionary significance of such analyses have always eluded me, I'm content with keeping them in the manuscript. However, I find it a bit unfair to compare the Torix, which is a deeper clade and apparently in itself a grouping of the Limoniae, Leech and Moomin groups, with lower level groupings such as Bellii, SFG, Scapularis, TG, etc (see Figures 2 & 3). It would seem more fair to me to repeat the analysis but with the Torix clade broken up in its 3 constituent groups, which each have divergence levels comparable to the other Rickettsia groups.

427-428, Figure 8: The authors here suggest that the pentose phosphate pathway is likely an ancestral feature that was lost in the main Rickettsia clade. Please explain briefly how you came to this conclusion. I suspect its deduced from the presence in Megaira and Torix/Tisiphia and the general trend towards gene loss in Rickettsiales that the PPP must have been present in the last common ancestor of Megaira/Torix/Rickettsia. If feasible, the authors could do a phylogenetic analysis of key proteins in the PPP and check if Megaira and Torix representatives group as sister taxa in the resultant phylogeny. This would give some insight on whether this pathway was vertically inherited from a common ancestor, as the authors suggest, or perhaps independently acquired through horizontal gene transfer from some unrelated clade.

429-430, Figure 8: I'm a bit confused here. If I read Figure 8 correctly, it seems that Oopac6 also lacks glycolysis and gluconeogenesis pathways, contradicting the statement of the authors here. I'm checking the 3 rows under the highlighted PPP rows, but perhaps the authors are looking somewhere else? In any case, please highlight the part of Figure 8 relevant to this statement, similar to other highlighted areas. Also, it is unclear to me which rows in the figure are meant with "cofactor and vitamin metabolism". Perhaps the authors were referring to the biotin synthesis pathway discussed in the next sentence, but I found this a little confusing.

Figure 8: The NAD biosynthesis pathway is highlighted, it being uniquely present in the Moomin genome. It is however not discussed in the manuscript. It seems interesting, so please discuss it.

430-439: I assume that Oopac6 is considered a member of the Rickettsia genus. I do not understand why the authors claim that the biotin operon is absent in all other members of the Rickettsia genus when they themselves very clearly show in Figure S7 and in the text that Rickettsia buchneri also has this operon, with the same synteny no less. It was also unclear to me why the biotin synthesis pathway of Oopac6 is called "distinct" from the one in R. buchneri. Please elaborate

333-344: Please show the data/evidence that made you conclude the Oopac6 and Ppec13 genomes have low pseudogenisation, I could not find it. What do you mean with the "pangenome and metabolic profile"? No figure or table reference is given here. I also do not see how either a pangenome or a metabolic profile suggests that the Meloidae are a sister group to Belli. Perhaps you were referring to the phylogenomic tree in Figures 2 & 3 ? I assume that "this draft genome" refers to Ppec13, but it is not immediately clear, I would simply replace "this draft genome" with "Ppec13". I would also not use the word "linking" (l 342) in an evolutionary context as that may associate the readers mind with the popular phrase "missing link". Oopac6 is not a missing link between Torix and Rickettsia clades. I would simply state Oopac6 is sister to all other Rickettsia and leave it at that.

371-372: Please elaborate on what is meant with "neglected" symbiotic Rickettsiaceae (perhaps also include a reference here), which groups in particular? Also please explain very briefly with what is meant with an "open" pangenome. Finally, I'm guessing you meant to place a comma after "pangenomes"?

Figure 1: Please add which long and short read sequencing technologies were used.

Figure 2 and 3: We know that Orientia is the outgroup, why not simply show a rooted, ordered phylogeny? The current figures show as if the root is unresolved, which is unnecessary. An ordered, rooted tree can be easily done with Figtree.

383-384: I'm guessing Figure 2 (l 383) should be Figure 4, and Figure 4 (l 384) should be Figure 5?

166: The pangenome was constructed including Ca. Megaira and Torix Rickettsia, which are later assigned their own genus, Ca. Tisiphia. It is therefore incorrect to state that the pangenome was "constructed for Rickettsia"

Reviewer #1 (Remarks to the Author):

The manuscript “Large-scale comparative genomics unravels great genomic diversity across the Rickettsia and Ca. Megaira genera and identifies Torix group as an evolutionarily distinct clade” by Davidson et al. presents a massive analysis of dozens of novel rickettsial genomes that informs on the classification of genus Rickettsia and the evolutionary characteristics of the genomes of these obligate intracellular bacteria. This work comes off the heels of several papers by Dr. Hurst and colleagues analyzing metagenomic datasets (mostly insect genome sequencing projects) and other publicly available data to determine that a basal rickettsial lineage, termed “Torix Group Rickettsiae” is widespread in nature, particularly in non-bloodfeeding arthropods but also in some vectors and potential vectors of human disease. These works, combined with the present study, are critically important to human health because they not only enlighten on the factors leading to the highly pathogenic species (i.e. agents of typhus and spotted fevers) but they place Rickettsiae alongside Wolbachiae in as far as diversity and widespread occurrence in arthropod populations. Thus, as some have argued, approaches tailored for Wolbachiae for biocontrol of arthropod-borne pathogens are applicable to many Rickettsia species that seemingly persist in arthropod populations on par with their rickettsial cousins in the Anaplasmataceae.

There are several major accomplishments of this work that I point out below:

- 1) The first closed genomes of “Candidatus Megaira” (from the algae *Mesostigma viride*) and Torix Group Rickettsiae (from the midge *Culicoides impunctatus* and bed bug *Cimex lectularius*) are provided by the team. Closed genomes from these lineages are critical to understanding the Rickettsia pan genome and the extents of lateral gene transfer and metabolic capacity.
- 2) The team sequenced and constructed draft genomes for “Candidatus Megaira” from another algal host (*Carteria cerasiformis*), a Transitional Group Rickettsiae from tsetse fly (*Glossina morsitans morsitans*), and a Torix Group Rickettsiae from a spider mite (*Bryobia graminum*). These genomes are invaluable for conducting robust phylogenomic analyses as they will critical gaps in the diversity already in hand.
- 3) The team further extracted 22 draft genomes from arthropod genome sequencing projects, including tentative species from Adalia Group Rickettsiae (n = 1), Transitional Group Rickettsiae (n = 4), Spotted Fever Group Rickettsiae (n = 1), Torix Group Rickettsiae (n = 7), Belli Group Rickettsiae (n = 7), Rhyzobius Group Rickettsiae (n = 1) and Meloidae Group Rickettsiae (n = 1). Importantly, for the latter two groups, these are the first genomic datasets constructed to allow robust phylogenomic analysis.
- 4) Finally, the team effectively utilizes this massive genomic dataset of new and existing genomes to characterize the basal lineages previously named Torix Group Rickettsia and identify key attributes in the metabolome and accessory genome that, along with genetic divergence in estimated phylogenies, warrant removing this clade from genus Rickettsia. The team proposes the genus name “Tisiphia” as an immediate sister lineage to the remaining Rickettsia lineages.

These four accomplishments are probably the most important contributions to Rickettsia biology and evolution in the last decade. They stand to enlighten on specific works focusing on Rickettsia pathogenesis since we may now understand the origin and

maintenance of describe pathogenicity factors across a much broader and robust evolutionary framework. Furthermore, the completeness of the genomic datasets provides taxonomic resolution to the basal lineages that have been plagued by incompleteness and lack of thorough analyses.

The manuscript is also very well written and a pleasure to read.

I recommend acceptance of this monumental work after a careful revision that considers the following:

C1.1. General statement: usage of genus names as nouns is incorrect, though sadly commonplace. Genus names should be modifiers of either a species name (species epithet) or a noun (Wolbachia gene, Rickettsia phylogeny, etc.). “Genus Rickettsia”, “Rickettsia species”, “Rickettsiae”, etc. are all correct. All provisional names should be used as follows: “Candidatus <Non-italicized genus name> <non-italicized species name>”, or with “Candidatus” abbreviated to “Ca.” but still in italics.

We feel that the first point is a matter of writing style preference and defer to the decision of the journal editors. We would argue that Rickettsia is a name and therefore a noun, and its use is similar to previously published papers.

All mentions of Megaira that are missing a Ca. in the figures have been fixed, thanks for spotting that!

C1.2. Line 64: italicize Alphaproteobacteria.

Taxonomic names above genus level are traditionally not italicised. We will defer to the journals formatting preferences for this.

C1.3. Line 71-77: perhaps here or elsewhere, Gillespie et al. (PMID: 25477419) identified a plasmid named pLbAR that carries a toxin-antidote (TA) module purported to distinguish booklouse-associated Rickettsia felis from flea-associated R. felis; the TA module was later shown to carry the domains of Wolbachia CI factors and hypothesized to underpin parthenogenesis induction in booklice, which are all-female when infected with R. felis (PMID: 30060072).

This is referred to in the discussion (lines 272-275).

C1.4. Line 78: Only two years prior Gillespie et al. (PMID: 17342200) analyzed existing genomes, particularly plasmid pRF from R. felis, and concluded that a lineage distinct from Typhus Group Rickettsiae and Spotted Fever Group Rickettsiae should be recognized. This clade, termed Transitional Group Rickettsiae, has since been well separated from the Spotted Fever Group Rickettsiae with the availability for more genomes as well as more sophisticated phylogeny estimation tools. These two factors alone have the potential to lead to continual reorganizations of Rickettsia classification as the authors demonstrate here.

This is now cited (line 59) with the most recent revisions of rickettsia phylogeny.

C1.5. Line 78-100: per my comment above, designations of “Groups” based on a few taxa and a few genes is pretty tenuous. How many of the 13 groups in Weinert et al. 2009 will hold up when more species and genome sequences are unearthed? Is there a negative impact on the field when these “Groups” are proposed but later have to be revised? Also, some of the groups names are really superficial, describing one type of host for a few members that have different hosts (e.g., the “Ixodes Group” comprises R. tamurae and R.

colombianensi that infect mostly Amblyomma ticks, and R. helvetica can infect Ixodes ticks but is not within this clade!).

See also above. We also agree that Rickettsia groups can be superficial in nature as some of them rely on limited data, however they are a convenient means of separating like species. Here we used the traditional grouping nomenclature to help readers navigate and compare with the previously published literature. The aim of this paper is not to redefine Rickettsial taxonomy (aside from Torix/Tisiphia). If enough genomic data becomes available, maybe it will be possible to formally define species within non-SFG Rickettsias (possible interesting future project?)

C1.6. Line 101: this idea has been proposed before (PMID: 25073875 and PMID: 23475938) and may have been touched on by Perlman et al. in their seminal report (PMID: 16901827).

This paragraph was written assuming wider knowledge and was not properly referenced in general. The whole section has been more appropriately referenced in the revised manuscript (line 84-87).

C1.7. Figure 1: very nice flow chart. Some of the text is difficult to read.

Figure 1 has been amended to enhance readability. The full size image is also available on figshare and is linked to in the legend.

C1.8. Line 129: having A and B sections here without panels in Figure 1 is a little confusing.

Done

C1.9. Line 134: the reader is introduced to Orientia tsutsugamushi for the first time here. It might be worth introducing this species in the Introduction...it is relevant that it was once called Rickettsia tsutsugamushi and now at this moment is so far removed relative to all of this new diversity. Some readers could benefit from this information.

This is interesting information for the history of Rickettsia, however, this paper is not about Orientia or the history of Rickettsia clade. We have instead cited Tamura et al. 1995 (line 121).

C1.10. Lines 143-156: Are all materials deposited as vouchers? Is the genetic material archived and available?

Raw genomic data is available on NCBI database.

Cimex lectularius is from a currently maintained lab colony, as is Moomin. Carteria cerasiformis strain NIES 425 is a voucher specimen from the Microbial Culture Collection at the National Institute for Environmental Studies, Japan. Glossinia morsitans submorsitans material was field collected for previous work several years ago, the source material was not kept. Culicoides impunctatus is readily available from the Scottish highlands and efforts are being made by Jack Pilgrim to maintain it in culture, but so far is not breeding.

C1.11. Line 211: the specific supplementary figure with the ribosomal protein tree is not called out.

Link added

The Materials and Methods are very well described and easy to follow, with all appropriate references provided. It is conceivable that all methods followed as described would lead to similar results obtained by the authors.

C1.12. Line 241: This is an interesting finding. Gillespie et al. identified only ONE gene that is present on all Rickettsia plasmids: an odd DnaA domain-containing protein (PMID: 25477419). Is this present on these plasmids? Are there RAGE genes that tend to be on most of the Rickettsia plasmids? It would be interesting to learn your opinion on the origin of these plasmids and if they provide any links between Tisiphia and Rickettsia species, or exchanges with other microbes.

Yes both tisiphia plasmids, UCM86444.1 (pRiCimp002) and UCM86407.1 (pRiCimp001) carry distant homologues of this DnaA_N like protein. In addition, RAGE genes are also present including the tra genes traD and traA. The Rickettsiales specific RAGE elements alongside the fact conjugation apparatuses have narrow host-ranges suggest HGT of these plasmids is likely occurring within the Rickettsiaceae and likely between Tisiphia and Rickettsia considering co-infections of these genera are being seen more and more frequently. This information has now been included on lines 232-246.

C1.13. Line 261: fascinating!

C1.14. Figure S3: it would be helpful to know the top blastp hits to these interesting proteins. Are they similar to the R. buchneri cassettes or do they have a different evolutionary profile? Perhaps a little digging in this regard might enlighten on a possible function? Also, are they syntenic with any other microbes? You could just blastn the entire nucleotide sequence for the cassettes and see it right away (or tblastx if the closest syntenic counterparts are too divergent at the nt level).

The taxonomic distribution of the top BLASTp hits for RiCimp and MegNEIS296 NRPS proteins are now included in the supplementary figure S3. Although the NRPS/PKS in these taxa have not been linked with any specific phenotypes, the Norine results already presented suggest toxin or antimicrobial peptides are likely candidates for the product of these.

C1.15. Figure S3: these images are pretty but hard to read...you could utilize the space better and make it easier on the reader.

The key in this figure has been made larger, as well as the co-ordinates on the circular chromosomes. The epimerization domains have been coloured orange to distinguish from its previously blue counterpart.

C1.16. Line 278: word usage suggestion here, "The Transitional Rickettsia" could be written "the TRG Rickettsia species".

Revised

C1.17. Line 278: This is interesting, but I wonder if this information will be associated with the data on NCBI and other databases? Or will others that don't read this work have to suffer the consequences. Or is it that these are so close (strains I assume) that it doesn't really matter for tree-building and such. Are there more than one 16S rDNA sequences?

We appreciate the comment. We have notified GenBank to include a note on this matter. Unfortunately, we could not unambiguously separate the potential variants with the tools that we have, they are extremely similar (~0.2% of biallelic sites were identified) and are

unlikely to affect analyses. There is only one 16S sequence (hmms hits, identified with anvio).

C1.18. Line 286: How were Lappe3 and Lappe4 unambiguously assembled?

The average nucleotide similarity between the two genomes is ~91% so they are quite distinct and can be regarded as different species. This makes it easier to separate the assemblies based on tetranucleotide frequencies and contig depth of coverage. We used Metabt2 to cluster the contigs into separate bins. Subsequent QC using CheckM further shows limited contamination far below the acceptable levels for a high quality MAG (0.32% for Lappe3 and 1.18% for Lappe4; the accepted boundary for good quality is <5%)

C1.19. Table 2: great care was taken to assess the relative completeness of the existing Rickettsia genome assemblies; how well are these new assemblies in relation to the existing ones? How often does each new assembly disrupt a core gene set? Is there some metric that can be used to assess and rank the relative completeness of these assemblies?

*Contamination and completeness information is in supplementary table S1
<https://doi.org/10.6084/m9.figshare.14865561>*

In response to this comment we have altered the legend for table 2 to elaborate on the metadata available in S1.

Completeness and contamination were assessed with checkM which compares bacterial genomes against a set of marker genes (<https://ecogenomics.github.io/CheckM/>). All of the new genomes score between 94% to 100% completeness. Contamination is below 1.66% for all except the two megaira which have scores of 3.32%. Scores below 5% are generally accepted as good in bacterial metagenomic assemblies.

CheckM was also used to assess the completeness of existing rickettsia genomes used in all of the phylogenomics in this paper. Several existing genomes actually scored quite poorly and were not used in further analyses (also detailed in S1)

Re: “How often does each new assembly disrupt a core gene set?” disruption was minimised as much as possible with quality checks (as described in the methods), this resulted in the removal of previously existing poorer quality genomes to improve the core gene set.

C1.20. Line 307: The original paper on the seal fur louse Rickettsia species should probably be cited here.

Agreed, now cited! (Was previously listed in S1 as GCA_001602635.1)

C1.21. Line 310: You use “Transitional” as if it is an adjective but follow with “Spotted Fever Group”. It should be TRG or Transitional Group Rickettsiae.

Removed “group” to be more consistent with labelling in figures and made it clearer that all of the names listed are group names (line 323-325)

C1.22. Line 316: Guilotte et al. recently showed R. helvetica as basal to TRG, TG, and SFG Rickettsiae (Figure 2 in PMID: 33952661). This tree was modified from Hagen et al. that also reported the same phylogenetic position for R. helvetica (PMID: 30398619).

See response on the comment below.

C1.23. Line 316: The text here is confusing...why would you posit *R. helvetica* is most similar to the “Scapularis group” if it belongs in a unique clade well removed from this group? Furthermore, including only one member of the “Scapularis group” in your phylogeny estimates makes it seem as if this clade is not stable. There are plenty of conserved genes for the *I. pacificus* and *I. scapularis* endosymbionts to provide stability and show that *R. helvetica* does not belong to this clade. Do you have some gene profile support for your supposition?

We agree that these conclusions might be premature given that only one genome was used for both Scapularis and Helvetica groups. This observation was mainly supported by our previous phylogenies which included more scapularis genomes, however, all but one scapularis failed the relatively strict QC parameters for further use (contamination <2% or strain heterogeneity <50%). All of these unused genomes are available in S1. We decided to remove this section as it doesn't add to the main focus of the manuscript and will rather create more confusion.

C1.24. Figures 2 and S4: Two problems here. One, the divergence can hardly be seen for most clades. This could be solved by collapsing all monophyletic strains (e.g., *R. prowazekii* and *R. rickettsii*) and truncating the species names (i.e. *R.* instead of *Rickettsia*) so the figure can be expanded. Two, a simple cladogram can be shown to the right of clades with little divergence. Otherwise, these trees will remain difficult to read.

Monophyletic strains have been collapsed as suggested, and the whole tree expanded horizontally to aid readability.

C1.25. Figures 2 and S4: It would cool and perhaps informative to map the clades that differ across estimations. Does the discordance jive with low support values?

We did not detect significant discordance between the ribosomal and core phylogenies. Now mentioned (lines 318-319).

C1.26. General comment: what are your criteria for naming groups? Monophyly? A certain degree of divergence on estimated phylogenies? Can a group be less than two entities? Is “*Canadensis*” a group if only two strains form the clade?

*Please refer also to our response on comment C1.5. The use of groups is based on the existing nomenclature and used for reference purposes. Our aim is not to redefine *Rickettsia* groups.*

C1.27. Figure 3: I am not really sure of the value of this analysis in light of the phylogeny estimation; so few genes are analyzed and they should be stable and conserved. Obviously, such a network will get messy when less conserved genes are analyzed.

We appreciated the comment. We have moved this Figure in the Supplementary material.

C1.28. Line 333: What is the history of “*Rhizobius* Group” and why is it important to keep this naming system? Could the “*Meloidae* Group” possibly be combined with the *Bellii* Group and given one name? Or is it too divergent for that? It seems that the *Onychiurus sinensis* associated *Rickettsia* species may indicate further group resolution down the road.

*Rhizobius is what it was originally called because the first instance of this *Rickettsia* was found in *Rhizobius litura*. We kept the naming system that was already in place to reflect previous nomenclature.*

Yes, Meloidae group could be potentially combined with the Belli group. However, as above we wanted to reflect the previous nomenclature.

Onychiurus is also defined as another group but only based on one 16S sequence and does not have a whole genome to allow definitive inference. We agree that it is likely another group.

C1.29. Figure S5: this tree is difficult to read like the others (can some of the close divergences be better illustrated?). It seems also that the trees are not ordered...at first glimpse it looks confusing and contradictory to the trees in Figures 2 and S4. I think the only glaring difference (monophyletic TG + SFG) would emerge better with ordering and showing the divergences better.

This is purely a result of branch orientation. Now figure S6, the tree has been rotated to match the clade order in Figures 2 and S4.

C1.30. Line 354: There are a lot of others, most reviewed in PMID: 23475938, which also provided scaffold and transcriptional evidence for rickettsial genes in the *Trichoplax adhaerens* genome. More recently, a *Wolbachia* CI antidote was shown inserted as an exon in a larger cat flea gene (PMID: 33362982); the CI genes themselves are often found in eukaryotic genomes (PMID: 30060072). Not all of these are *Wolbachia*-like...some are *Rickettsia*- and *Cardinium*-like!

Text has been revised to be more specific (365-366).

C1.31. Line 371: revise English for clarity.

The Pangenome section has been edited for clarity and to reflect the revised figures 4&5. An additional paragraph discussing features of the accessory genome is also included in the revised manuscript (section starting line 378)

C1.32. Line 379: what features distinguish this accessory genome?

Please refer to our response on the previous comment.

C1.33. Figure 4: this is difficult to read...there seems to be room to enlarge the taxa at the top right. Also, some metric would be nice to associate with the sizes of the accessory genomes per group (averages?). The arrangement of the rings seems strange...why are the groups out of phylogenetic order (radiating from center)?

*Accessory genome size depends on the number of genomes in each group. Following the request from two reviewers (please see C2.1 and C3.1) the pangenome analysis has been re-worked to reflect a genus level comparison between *Ca. Tisiphia* and *Rickettsia* rather than *Tisiphia* and individual *Rickettsia* groups. The updated rarefaction and upset plots that aim to provide some insight into the accessory genomes of *Ca. Tisiphia* and *Rickettsia*.*

The rings are ordered by gene cluster frequency, not phylogeny.

Annotation on the figure has been rearranged to aid legibility, and the whole figure has been moved to the supplementary based on suggestion from other reviewers.

C1.34. Figure 5: this could be a supplement to save space; it is sort of implied from the phylogeny estimations and is in agreement.

The ANI/AAI is important for species and genus definition for uncultured microorganisms. We have now remade this as a Network showing species- and genus-level clustering based on ANI and AAI values. We feel this is a much clearer representation of what we were trying to convey.

C1.35. Figure 6: it would seem more useful to me if the taxa were arranged phylogenetically at bottom rather than by cluster size. It is also difficult to read a lot of the text in this figure. It seems better arrangement and space minimization could make things clearer, larger font sizes too.

This figure (now figure 4) has now been reworked and simplified from feedback from multiple reviewers. It now compares genomes across Rickettsia, Rhizobius, Torix and Megaira to make more appropriate comparisons across genus-level groups rather than mixing them.

Figure 7: this is a very cool figure and I am glad to see the permutations conducted. Well done!

Thank you. This figure (now figure 5) has been reworked to reflect a genus-level comparison between the main Rickettsia clade and Torix (Ca. Tishiphia).

C1.36. Line 427: This was concluded by Driscoll et al. (PMID: 28951473) and can be inferred from comparisons with sister Rickettsiales lineages. It also can be explained by the presence and ability of Rickettsiaceae to import ribonucleotides required for interconversions to deoxyribonucleotides.

Also in doi: 10.1111/1462-2920.13887 , both are now cited (line 449)

C1.37. Line 433: I have tried hard to understand this sentence: “Based on the gene cluster comparison plot and an independent blastx search, the GlyA gene in Rickettsia buchneri appears to be a misidentified bioF gene”. Is this some different annotation that was reported by Gillespie et al. a decade ago? Please clarify. NCBI likes to turn wine to water when it comes to annotations.

We retrieved the annotated contig containing the R.buchneri biotin operon from GenBank and it seems this is a case of mislabelling BioF as “GlyA” The sentence has been removed to avoid confusion. The supplementary figure has been corrected.

C1.38. Line 435: I don’t understand this sentence either: “Additionally, the insect SRA sample was not infected with Wolbachia, making it unlikely that the presence of the biotin operon is a result of misassembly”. Why bring up Wolbachia here? The figure shows greatest similarity between Oopac6 and R. buchneri BOOM, so what does Wolbachia contamination have to do with anything?

Agreed that this is unnecessary, removed for clarity

C1.39. Line 438: “Oopac6 has retained or acquired a complete biotin operon where this operon is absent in other members of the genus”; you need to estimate a phylogeny like Driscoll et al. recently did (Figure 2C in PMID: 33362982) and determine this. Based on the similarity between Oopac6 and R. buchneri, it is likely the BOOM invaded Rickettsia species multiple times. The loss of bioH in Oopac6 is telling; have you looked for it elsewhere in the genome or identified any other non-orthogonal methyl esterases? There are several different kinds that bacteria use to regulate biotin synthesis (see Figure S4 in PMID: 33362982).

We believe Figure S7 sufficiently illustrates the strong similarity of Oopac6 to the rickettsia biotin. Due to the limited number of biotin cluster from Rickettsia and Ca. Tisiphia a phylogeny will not provide more detail. We agree that the biotin clusters might introduced independently multiple times but we can not exclude the possibility that it was ancestrally present/introduced in the Rickettsia-Ooac6 clade.

Pimeloyl-ACP biosynthesis seems to be at least partially present across all Rickettsia, but BioH does not appear inside or outside the biotin operon as far as we can tell. Blast searches did not reveal distant homologues of bioH either.

Have added in lines 468-472

See also C1.43

C1.40. Line 445: The synteny of the BOOM is telling in relation to other biotin synthesis gene operons and clusters. It would seem strange that the synteny would be similar to BOOM if Oopac6 had biotin synthesis capability and secondarily lost it. There is no evidence for an alternative bio gene arrangement for comparison. It seems more like that Oopac6 picked it up and maybe is in the process of losing it (loss of bioH); it could also be a symbiont that is experience a recent host shift and no longer benefits from supplying biotin to a host that gets plenty of it from its diet.

See C1.39 and C.43

C1.41. Figure 8: very difficult to read. Some order of the metabolic processes would help as well.

All specific metabolic pathways are listed in the supplementary and full resolution image can be accessed on figshare this has now been added into the legend. Unfortunately because it is so large we are limited to it having very small text (a common problem in these types of data visualisation). We have done our best to alleviate this by increasing font sizes and the width of heatmap cells. The ordering of each process is assigned according to Pheatmap clustering, meaning they are listed from most universal to least universal presence.

C1.42. Line 467: This is interesting. It could mean that loss of rhamnose in the O-antigen is more of a characteristic of hematophagous host-associated species.

Possibly but I don't think we can conclude that from the data we have since this pathway is also absent in non-hematophagous species.

C1.43. Line 496: Is the pathway complete in the absence of bioH or another methyl esterase?

Pimeloyl-ACP biosynthesis (M00572) seems to be at least partially present across Rickettsia, including Oopac according to metabolic predictions. However without actually testing functionality of the gene, we cannot know for certain what these genes do, if they still work.

See also C1.39

C1.44. Line 497: Agreed on all counts. This is a fabulous contribution to Rickettsiology and will have a tremendous and lasting impact. A massive effort. Kudos to the authors.

Thanks!

Reviewer #2 (Remarks to the Author):

In the manuscript 'Large-scale comparative genomics unravels great genomic diversity across the *Rickettsia* and *Ca. Megaira* genera and identifies *Torix* group as an evolutionarily distinct clade' Davison and colleagues present novel *Rickettsiaceae* genomes, both from specific sequencing efforts and database mining. The genomes, that highly enrich the known genomic diversity of *Rickettsiaceae* are analyzed through phylogenetics and comparative genomics methods. The authors convincingly conclude that the *Torix* group is different enough to merit genus status. The work is of interest and provides a significant advance in the field.

The methods used are generally solid and well explained. The sequencing is performed with both long and short reads (on some samples) allowing to get complete genomes. Comparative genomics is performed with multiple tools allowing to compare synteny, gene content, nucleotide identity and 'functional' annotation. The manus is well written and clear, with very few minor typos (see below). The quality of the supplementary materials text appears a bit lower, and could use some polishing. The main figures are in general clear and vehiculate the message, see below for specific comment. The supplementary figures and materials are informative.

The work, however, presents one main issue (or multiple connected issues).

C2.1. Considering that the authors delineate *Torix* as a new genus (and I agree with this) and show the presence of subgroups in *Torix*, I would find appropriate and interesting to compare the diversity of classical *rickettsia* groups to the *Torix* groups, not to *Torix* as a whole. As they stand now, the comparative analyses have a little bit of an 'apple to oranges' problem. Either compare true *Rickettsia* to *Tisiphia* as a whole, genus to genus, or *Rickettsia* groups to *Tisiphia* groups (as let's say species to species). The ANI of the 'true *rickettsia*' groups is much higher than the novel ones, clearly highlighting this issue.

*We agree with the reviewer that the comparison as it stands now may not be appropriate considering that *Torix* group can identified as an separate genus. However, we also believe that a comparison between *Rickettsia* groups and *Tisiphia* groups may not be appropriate either for the following reasons: a) current grouping within *Tisiphia* but also *Rickettsia* is rather arbitrary and does not represent species level grouping. For example, based on ANI values and GTDB classification *Tisiphia* may consist of 6 putative species. b) most of the *Tisiphia* groups are under-represented which makes analysis questionable. We have updated our results with a genus to genus comparison between true *Rickettsia* and *Ca. Tisiphia* (*Torix*). The taxonomic status of *Rhizobium* is a bit unclear and was excluded from the true *Rickettsia* genomes as it may consist a separate genus itself (GTDB taxonomic status outside *Rickettsia* genus [see table S1]). Networks for ANI and AAI have now been made, which make it clearer how genera and species groups delineate in comparison to traditional grouping conventions. This replaces the ANI heatmap.*

C2.2. Related to this is the pangenome analysis. First I feel it makes no sense to evaluate the pangenome of the entire dataset (which includes *Megaira* and *Orientia* if I correctly understand). The genomes are too diverse because the sampling is too broad, being almost the pangenome of the entire *Rickettsiaceae* family. Secondly, we have again the 'apple to oranges' problem. Of course the pangenome of *Tisiphia* is wider than that of

Rickettsia group, because it is that of an entire genus. While I understand this fortifies the notion that Tisiphia is a genus, then it leads to the very strong conclusion that the pangenome is open. An open pangenome implies a higher genomic plasticity, which would be a huge finding for a group of intracellular symbionts, but is this the case? Is it a true result or is it due to the higher diversity of the novel groups?

If the analyses were performed lumping all the true Rickettsia together, would the authors find an open pangenome? If this was the case, would it be a true biological result? I would argue it would not. Pangenomes should be of species, not of genera. I suggest to either strongly modify this analysis or to remove it completely.

We appreciate this comments. In the revised version of the manuscript we have re-worked the pangenome and gene content comparison section to better reflect a genus level comparison between Rickettsia and Ca. Tisiphia. Upsetplots and accumulation plots have now been revised in response to this and other concerns. They now compare at the genus level rather than between arbitrary traditional groupings. We believe this still reflects greater genomic plasticity within Torix/Tisiphia compared to traditional Rickettsia.

Orientia has been removed as suggested from the pangenome figure and the upsetplot plot, it is used only as an outgroup for phylogenomic comparisons.

Megaira is included in the gene content analyses for comparative reasons as it was previously considered a Rickettsia group (the Hydra group) until very recently and now has its own genus status; Megaira is not included in the gene cluster accumulation analyses as it only has two representative genomes.

C2.3. Related to this, is the issue of Megaira diversity. Is it really lower than Torix diversity (as it is somewhat implied for example at line 377), or is it only perceived so due to the lack of genomic resources? Previous trees of Megaira diversity (16S based) would suggest the latter. The authors should address this in the text.

We agree with the reviewer. The diversity of Megaira is underestimated due to limited taxon sampling. In fact if we consider the number of genomes available per clade the number of the Megaira specific orthologous clusters per genome outnumbers both Torix and Rickettsia clades suggesting even higher diversity in Megaira. A warning has been added in text that diversity is underestimated in these clades with less data (line 394-395)

Additional issues are highlighted below.

C2.4. A comment/suggestion on the tone. The manuscript contains the word 'first' 19 times. I did not check them all, but a good number of them are used to indicate the novelty of the results. I would reduce this number a bit. While the novel genomes are the foundations of this work, I would try to move the emphasis balance more towards how they were analyzed and what the authors learnt from them. Saying 'this is the first' a couple of times less would not make the results any less novel but would make the manus sound, in my humble opinion, a little bit classier.

Thank you for flagging this, we have reduced the instances of "first" considerably.

C2.5. Gene content analysis: figure 8 shows four main findings. Three are discussed in too much detail in the text, one not enough. Please reduce this part, and make it more balanced, adding a sentence on Nad synthesis (e.g. present in doi: 10.1186/1944-3277-9-9 and doi: 10.1111/1462-2920.15396).

Done line 495-501

C2.6. Line 84, 106...: Megaira is not the sister group of Rickettsia. The authors completely ignore Trichorickettsia and Gigarickettsia. No genomes are available for these two genera, but their existence should be at least acknowledged.

Reworded to "related groups" avoid confusion about terminology.

We have added mention of these other rickettsiales in the introduction (lines 84-87)

C2.7. Line 246: I find this result to be really interesting, especially for how extreme the lack of synteny is (Fig S3). A short discussion on this result would be nice, for example comparing it to previous work by one of the authors on Wolbachia (doi:10.1098/rsob.150099.) or discussing how common this lack of synteny could be in Rickettsiales. Maybe the generally accepted notion of symbiont genomes plasticity to be all about losing genes, and the low number of complete Rickettsiales genomes, have led to underestimating this trend until now?

This could be discussed in connection with the pangenome results (if updated).

We appreciate the comment. We have revised the paragraph to incorporate the relevant literature (lines 252-258). We believe that the extreme lack of synteny between RiCimp and RiClec is due to a combination of their evolutionary distance and the proliferation of mobile elements as previously suggested for both Wolbachia and Rickettsia. We agree with the reviewer that this genomic plasticity might be more common, however, a thorough analysis of the phenomenon in the Rickettsiales is beyond the scope of this paper and we feel that it would require more data to be meaningfully interpreted.

C2.8. The main tree (fig 2). Iqtree bootstraps should be color-coded differently, dividing 91-100% and 81-90%, as in IQTREE these values are considered differently (>90% equals high support).

Done

C2.9. Thypus and spotted fever are highly overrepresented.

Could this lead to errors in the phylogenetic reconstructions? Unlikely, but worth checking. I would like to see the same analyses with a smaller, more balanced dataset (e.g remove multiples from same species).

The trees from the complete datasets could be moved to the supplementary materials, and the smaller tree would make better main figure, with a stronger focus on the novel groups.

Also the tree should be expanded horizontally, to allow to better gauge branch lengths.

The revised trees has been expanded horizontally to aid legibility of branches and large branches of the monophyletic species have been collapsed to improve the focus (e.g Rickettsia conori, R. prowazaki etc.)

We did smaller trees very early on because we had the same concerns and there is no difference. Additionally, IQTREE by default removes identical sequences during tree construction and places them after making the tree (which is very nice). Indeed some of the SFG and TG datasets were flagged as identical by iqtree but is unlikely that this might have caused instability in our focal strains.

C2.10. Figure 4, while beautiful, is not very informative. Since there are a total of 8 main figures I would move this to supplementary.

Agreed, has been relegated to the supplementary material.

C2.11. Could Moomin and Rhizobius be different genera? I guess the authors think not (and I feel I agree), but based on trees and ANI, the authors should at least consider the possibility, and explain why they think they do not.

Yes Rhyzobius could very well be its own genus something that is also suggested by the GTDB taxonomic classifier. However, clustering based on AAI placed it with the rest of the Rickettsia genomes so there is some degree of uncertainty here. We chose not to explicitly state our thoughts on Rhyzobius due to this uncertainty, deliberately leaving it ambiguous (line 352-353)

On the other hand Moomin despite being an incredibly diverse clade, has been classified as member of the Tisiphia group (Both GTDB classification and our own AAI clustering agree). We have reworded the results to indicate that moomin is very different to other torix/Tisiphia (lines 325-326).

MINOR COMMENTS

C2.12. Line 64 and below: Rickettsiales and Alphaproteobacteria should be in italics. So Rickettsiaceae etc. Candidatus should be in quotes.
Taxonomic names above genus level are traditionally not italicised. We will defer to the journals formatting preferences for this.

C2.13. Line 64: there is one non-intracellular Rickettsiales (doi: 10.1038/s41396-019-0433-9)
Thanks for the reminder!! Now cited (line 43)

C2.14. Line 74: 'with the symbiont capable' clarify to something like 'with the different symbionts being capable...'
Changed

C2.15. Line 102: 'infect invertebrate symbionts' remove symbionts
Changed

C2.16. Line 211: please provide number for the SUPPLEMENTARY FIG.
Changed

C2.17. Line 307: add citations
Added

C2.18. Line 346: correct multilocus
Changed

Reviewer #3 (Remarks to the Author):

Davison et al greatly increase the sample of Rickettsia-related genomes by directly sequencing HMW DNA extracted from a range of arthropods and an alga, re-assembling data from a previously published genome sequencing effort of the alga *Mesostigma viride* and mining 29 publicly available arthropod SRA datasets known to contain traces of Rickettsia DNA.

All in all the authors manage to assemble 3 complete genomes and another 25 draft genomes. This awesome effort was particularly successful in reconstructing genomes from previously underrepresented Rickettsia clades. These include first genomes of the Meloidae, Rhyzobius and Megaira clades, and many additional genomes of the Limoniae, Leech, Moomin (collectively "Torix Rickettsia") and Belli groups.

In-depth comparative and phylogenetic analyses reveal the Torix clade to harbor a relatively large divergence and to be a sister clade of the Rickettsia genus. The authors propose a new name, *Ca. Tisiphia* for this group, a novel genus. *Ca. Tisiphia* and *Ca. Megaira* genomes were found to encode an intact pentose phosphate pathway, a feature lost in all other Rickettsia genomes. "Oopac6", the first genome of the Rhyzobius group, was found to encode a complete biotin synthesis pathway that is otherwise absent in the Rickettsia genus. Finally, four novel Belli group genomes were found to encode a near complete dTDP-L-rhamnose biosynthesis pathway, a unique feature among all Rickettsia genomes considered in this study.

I'm very impressed with the amount and the quality of the work, and the manuscript is generally well and clearly written. It is a substantial step forward into the field. I have learned a lot! I have no major comments. I do however, have several minor comments. Approximately from most-to-least important:

C3.1. 374-381, Figure 6 & 7: Here the Torix Rickettsia are compared with other Rickettsia groups in terms of gene content overlap with other groups and pangenome rarefaction curves. The authors conclude that out of all the clades, the Torix group has the most unique genes and has likely the largest pangenome (if I understand line 379 and Figure 7 correctly). Though the biological and/or evolutionary significance of such analyses have always eluded me, I'm content with keeping them in the manuscript. However, I find it a bit unfair to compare the Torix, which is a deeper clade and apparently in itself a grouping of the Limoniae, Leech and Moomin groups, with lower level groupings such as Bellii, SFG, Scapularis, TG, etc (see Figures 2 & 3). It would seem more fair to me to repeat the analysis but with the Torix clade broken up in its 3 constituent groups, which each have divergence levels comparable to the other Rickettsia groups.

Please also refer to our responses on comments C2.1 & C2.2 of the second reviewer. Now figure 4 and 5 have been updated to compare genus level clusters rather than groups. The relevant section in the manuscript have been revised (section starting line 378)

C3.2. 427-428, Figure 8: The authors here suggest that the pentose phosphate pathway is likely an ancestral feature that was lost in the main Rickettsia clade. Please explain briefly how you came to this conclusion. I suspect its deduced from the presence in Megaira and Torix/Tisiphia and the general trend towards gene loss in Rickettsiales that the PPP must have been present in the last common ancestor of Megaira/Torix/Rickettsia. If feasible, the authors could do a phylogenetic analysis of key proteins in the PPP and check if Megaira and Torix representatives group as sister taxa in the resultant phylogeny. This would give some insight on whether this pathway was vertically inherited from a common ancestor, as

the authors suggest, or perhaps independently acquired through horizontal gene transfer from some unrelated clade.

*We have previously performed phylogenetic analyses on key PPP proteins, see figure 4 and supplementary figure S5 in <https://www.ncbi.nlm.nih.gov/pmc/articles/PMC5656822/>. Based on these analyses and the presence of homolog proteins in *Occidentia massiliensis*, a sister species to *Orientia*, but also *Ca. Arcanobacter lacustris*, a deep Rickettsiaceae lineage we concluded that PPP was likely ancestrally present in the clade. More recently, a study on the genome sequencing of another member of the Rickettsiaceae (*Ca. Sarmatiella mevalonica*) and sister clade of both *Rickettsia* and *Megaira* also reported the presence of the non-oxidative phase of the PPP which further supports our previous conclusions.*

C3.3. 429-430, Figure 8: I'm a bit confused here. If I read Figure 8 correctly, it seems that *Oopac6* also lacks glycolysis and gluconeogenesis pathways, contradicting the statement of the authors here. I'm checking the 3 rows under the highlighted PPP rows, but perhaps the authors are looking somewhere else? In any case, please highlight the part of Figure 8 relevant to this statement, similar to other highlighted areas. Also, it is unclear to me which rows in the figure are meant with "cofactor and vitamin metabolism". Perhaps the authors were referring to the biotin synthesis pathway discussed in the next sentence, but I found this a little confusing. *Oopac does not have PPP, only the biotin synthesis pathway (B in figure 8). Reworded to avoid confusion. The section labelled "cofactor and vitamin biosynthesis" on figure 8 is for "cofactor and vitamin metabolism", reworded for clarity (lines 450-455)*

C3.4. Figure 8: The NAD biosynthesis pathway is highlighted, it being uniquely present in the *Moomin* genome. It is however not discussed in the manuscript. It seems interesting, so please discuss it.

Please refer to comment C2.5

C3.5. 430-439: I assume that *Oopac6* is considered a member of the *Rickettsia* genus. I do not understand why the authors claim that the biotin operon is absent in all other members of the *Rickettsia* genus when they themselves very clearly show in Figure S7 and in the text that *Rickettsia buchneri* also has this operon, with the same synteny no less. It was also unclear to me why the biotin synthesis pathway of *Oopac6* is called "distinct" from the one in *R. buchneri*. Please elaborate

*That should have been absent from all other Rickettsia genomes included in our phylogenetic analyses, apologies for the confusion! Now clarified (see answer to C3.3) It is distinct because it's similarity to *R. buchneri*'s is less than 95%. This clarification has been added in text (lines 450-455).*

C3.6. 333-344: Please show the data/evidence that made you conclude the *Oopac6* and *Ppec13* genomes have low pseudogenisation, I could not find it. What do you mean with the "pangenome and metabolic profile"? No figure or table reference is given here. I also do not see how either a pangenome or a metabolic profile suggests that the Meloidae are a sister group to Belli. Perhaps you were referring to the phylogenomic tree in Figures 2 & 3? I assume that "this draft genome" refers to *Ppec13*, but it is not immediately clear, I would simply replace "this draft genome" with "*Ppec13*". I would also not use the word "linking" (l 342) in an evolutionary context as that may associate the readers mind with the popular phrase "missing link". *Oopac6* is not a missing link between *Torix* and *Rickettsia* clades. I would simply state *Oopac6* is sister to all other *Rickettsia* and leave it at that. *Pseudogenisation was a typo and has been changed to contamination.*

*"Pangenome and metabolic profile" changed to phylogenies
Replaced "this draft genome" with "Ppec13". Also changed lines 351-353 to be clearer.
"Linking" changed to "of"*

C3.7. 371-372: Please elaborate on what is meant with "neglected" symbiotic Rickettsiaceae (perhaps also include a reference here), which groups in particular? Also please explain very briefly with what is meant with an "open" pangenome. Finally, I'm guessing you meant to place a comma after "pangenomes"?

This is a colloquialism that we have been using to refer to the non-tick, non-pathogenic species. The whole section has been re-worked in the revised version of the manuscript to address comments from this and other reviewers.

C3.8. Figure 1: Please add which long and short read sequencing technologies were used. Added to the legend as the figure would become overcrowded. Also included a link to the supplementary methods.

C3.9. Figure 2 and 3: We know that Orientia is the outgroup, why not simply show a rooted, ordered phylogeny? The current figures show as if the root is unresolved, which is unnecessary. An ordered, rooted tree can be easily done with Figtree.

Now rooted

C3.10. 383-384: I'm guessing Figure 2 (l 383) should be Figure 4, and Figure 4 (l 384) should be Figure 5?

Yes, changed!

C3.11. 166: The pangenome was constructed including Ca. Megaira and Torix Rickettsia, which are later assigned their own genus, Ca. Tisiphia. It is therefore incorrect to state that the pangenome was "constructed for Rickettsia"

Please see also our response to comment C2.1. The section starting line 378 has been entirely reworked

Reviewers' Comments:

Reviewer #1:

Remarks to the Author:

I thank the authors for addressing my concerns. Along with the other responses to the other reviewers, the revisions made to the manuscript have greatly improved an already impressive study. Nice work!

Reviewer #2:

Remarks to the Author:

The authors replied to all my comments, amending the manuscript and providing updated analyses. Some issues however still remain to be addressed.

Pangenome

The novel iteration of the analysis is more solid, but one of the problems remains.

Pangenomes of genera have been calculated and analyzed, but the inventors of the concept clearly meant it to be applied at the species level.

In simple terms, the pangenome concept is the realization that the genetic repertoire of a biological species, i.e. the pool of genetic material present across the organisms of the species, always exceeds each of the individual genomes and can be, in several cases, "unbounded": an open pangenome - Tettelin & Medini <https://doi.org/10.1007/978-3-030-38281-0>

I stand by my previous point that drawing conclusions about genome plasticity based on genus-based pangenomes is dangerous.

I ask the authors to clarify the limitations of their conclusions on this aspect (e.g. Line 395).

Names

I have never been a nomenclature extremist, but while surely many authors write bacterial names as they please, there are rules. I see no reason not to follow them.

Line 23: there seems to be something wrong in the structure of this sentence, please rephrase

Lines 293-294: why do the authors consider "likely" that the infection is coming from a recent blood meal?

Line 353: "The Rhyzobius-group symbiont is phylogenetically distant from most Rickettsia and is potentially a sister clade of Torix and the main Rickettsia clades" sister of which organisms? Torix+main Rickettsia together? Please clarify. Anyway, I am not sure what the authors mean here. The position of Rhyzobius is well supported, even in deeper nodes, in Fig 2 and S4 on fig S6, as sister to main Rickettsia. Clearly, this is somehow premature and being based on a single sample, and I fully agree that future analyses will be important to understand its precise position. However, I do not see why the authors seem to suggest a single potential alternative position (among many other possibilities), for which I wouldn't see any specific support in their data. Thus, I would suggest to stay more neutral and remove "and is potentially a sister clade of Torix and the main Rickettsia clades".

Line 453: I believe it is misleading to report that ". The Oopac6 biotin synthesis pathway is related to, but distinct from, the Rickettsia biotin pathway from Rickettsia buchneri (Gillespie et al., 2012) with which it shares less than 95% sequence similarity".

The sequence similarity is only slightly below 95% (is it at the nucleotide or amino acid level?), being 85-92%, depending on the genes. The only other difference is the presence of an additional hypothetical protein within the R. buchneri operon. Those conditions are (I think) compatible with a direct vertical inheritance of such genes, and this should be discussed. Also personally I found hard to follow S8 with only some selected pairwise sequence similarity among different operons, and if possible would suggest to provide information about other reciprocal pairwise comparisons.

Reviewer #3:

Remarks to the Author:

Most of the comments I had in my initial review report have been addressed. I found however still one minor issue:

l 351 - 352: "Phylogenies of Ppec13 suggest that (...) Rhyzobius sit as sister groups to Belli (...)"

l 353 - 354: "The Rhyzobius group symbiont is (...) potentially a sister clade of Torix and the main Rickettsia clades"

These phylogenies suggest that Rhyzobius is a sister group to all other Rickettsia genus taxa (i.e. Belli up to Spotted Fever). Not a sister group to Belli, nor a sister group to Torix, nor a sister group to Torix & Rickettsia.

REVIEWERS' COMMENTS

Reviewer #1 (Remarks to the Author):

I thank the authors for addressing my concerns. Along with the other responses to the other reviewers, the revisions made to the manuscript have greatly improved an already impressive study. Nice work!

Thank you for the comments and kind words!

Reviewer #2 (Remarks to the Author):

The authors replied to all my comments, amending the manuscript and providing updated analyses. Some issues however still remain to be addressed.

Pangenome

The novel iteration of the analysis is more solid, but one of the problems remains. Pangenomes of genera have been calculated and analyzed, but the inventors of the concept clearly meant it to be applied at the species level.

In simple terms, the pangenome concept is the realization that the genetic repertoire of a biological species, i.e. the pool of genetic material present across the organisms of the species, always exceeds each of the individual genomes and can be, in several cases, “unbounded”: an open pangenome - Tettelin & Medini <https://doi.org/10.1007/978-3-030->

38281-0

I stand by my previous point that drawing conclusions about genome plasticity based on genus-based pangenomes is dangerous.

I ask the authors to clarify the limitations of their conclusions on this aspect (e.g. Line 395).

We appreciate this comment. We agree with the reviewer that the pangenome concept was originally applied to better describe genomic diversity at species level. However, the same concept could be and has been applied to describe higher taxonomic clusters (<https://doi.org/10.1016/j.mib.2014.11.016>). We are aware of the potential issues arising when conducting pangenomic analyses of increasingly diverged group of sequences and we have amended lines (226-228) to clarify this issue. The purpose of our analyses was not to define or compare the pangenome size of Rickettsia and Torix clades since we know that this is not an easy task even at the species level. We rather use the observed trends of the gene accumulation analysis as a rough estimate of the genomic plasticity.

Names

I have never been a nomenclature extremist, but while surely many authors write bacterial names as they please, there are rules. I see no reason not to follow them.

Corrected according to editorial preference.

Line 23: there seems to be something wrong in the structure of this sentence, please rephrase

The Abstract has been revised according to editorial suggestions.

Lines 293-294: why do the authors consider “likely” that the infection is coming from a recent blood meal?

It seems to be a mix infection of different but closely related Rickettsia in the transitional group which are known to be blood borne and symbiotic. The high depth of coverage (104x) of the retrieved Rickettsia genome points to a symbiotic association echoing observations in this paper <https://doi.org/10.1016/j.cimid.2011.12.011>. We have amended the sentence to tone down the blood meal claim and made it clear that we think this could be an intracellular symbiont.

Line 353: “The Rhyzobius-group symbiont is phylogenetically distant from most Rickettsia and is potentially a sister clade of Torix and the main Rickettsia clades” sister of which organisms? Torix+main Rickettsia together? Please clarify. Anyway, I am not sure what the authors mean here. The position of Rhyzobius is well supported, even in deeper nodes, in Fig 2 and S4 on fig S6, as sister to main Rickettsia. Clearly, this is somehow premature and being based on a single sample, and I fully agree that future analyses will be important to understand its precise position. However, I do not see why the authors seem to suggest a single potential alternative position (among many other possibilities), for which

I wouldn't see any specific support in their data. Thus, I would suggest to stay more neutral and remove "and is potentially a sister clade of Torix and the main Rickettsia clades".

This was a missed miss-wording from the last round of revision, thank you for catching this. Amended (Lines 195-198)

Line 453: I believe it is misleading to report that ". The Oopac6 biotin synthesis pathway is related to, but distinct from, the Rickettsia biotin pathway from Rickettsia buchneri (Gillespie et al., 2012) with which it shares less than 95% sequence similarity". The sequence similarity is only slightly below 95% (is it at the nucleotide or amino acid level?), being 85-92%, depending on the genes. The only other difference is the presence of an additional hypothetical protein within the R. buchneri operon. Those conditions are (I think) compatible with a direct vertical inheritance of such genes, and this should be discussed. Also personally I found hard to follow S8 with only some selected pairwise sequence similarity among different operons, and if possible would suggest to provide information about other reciprocal pairwise comparisons.

The similarity scores reported by Clinker are at the amino acid level. We make this now clear in the figure and the Figure legend. Clinker by default displays an optimal order of the sequences based on hierarchical clustering of an all-vs-all similarity matrix (<https://academic.oup.com/bioinformatics/article/37/16/2473/6103786>). We make this now clear in the figure legend., all the other similarities are less than 70%. We have also added in figshare an interactive html to S8 where you can move all the components around and see how they compare.

Despite the sequence similarity we believe that the current data do not clearly support a direct vertical inheritance of the biotin operon for two main reasons: 1) The Rickettsia buchneri operon is located on a plasmid thus equally supporting a horizontal transfer scenario and 2) the two Rickettsia operons share no homology in the regions upstream and downstream of the biotin operon suggesting independent acquisition events. There is not enough data to conclude either way as it is not clear whether Oopac's operon reside on a plasmid or the main chromosome. However, the most parsimonious explanation points to a horizontal transfer scenario. We have added a sentence to clarify that HGT can't be ruled out. (lines 270-273).

Reviewer #3 (Remarks to the Author):

Most of the comments I had in my initial review report have been addressed. I found however still one minor issue:

I 351 - 352: "Phylogenies of Ppec13 suggest that (...) Rhyzobius sit as sister groups to Belli (...)"

I 353 - 354: "The Rhyzobius group symbiont is (...) potentially a sister clade of Torix and the

main Rickettsia clades"

These phylogenies suggest that Rhyzobius is a sister group to all other Rickettsia genus taxa (i.e. Belli up to Spotted Fever). Not a sister group to Belli, nor a sister group to Torix, nor a sister group to Torix & Rickettsia.

Must have missed this, apologies. Amended (Lines 195-198)